# LabBuilder: Protocol-Grounded 3D Layout Generation for Interactable and Safe Laboratory

Jianbao Cao [* 1 2] Zhangrui Zhao [* 1 3] Bohan Feng [* 1] Zixuan Hu [4] Rui Li [1] Haiyuan Wan [5] Chenxi Li [1]
Jingyuan Li [6] Wenzhe Cai [1] Lei Bai [1] Wanli Ouyang [1 7] Lingyu Duan [4] Di Huang [3] Minting Pan [1] Sha Zhang [7]
Xinzhu Ma [3] Shixiang Tang [† 7] Dongzhan Zhou [† 1]

## Abstract

Automated laboratories hold the promise of accelerating scientific discovery, yet their deployment is bottlenecked by the difficulty of designing safe and executable environments. While simulator-based design offers scalability, existing 3D scene generation methods are primarily tailored for household settings, optimizing for visual plausibility while neglecting the protocol grounding and layout-level safety constraints essential for scientific experimentation. We present **LabBuilder**, an end-to-end system that generates and verifies 3D laboratory layouts from concise textual specifications. It operates through three tightly coupled components: LabForge first curates a meta-dataset of annotated assets and chemical knowledge, translating natural language specifications into structured protocols; building on these protocols, LabGen synthesizes laboratory layouts via an iterative, constraint-aware optimization strategy; finally, LabTouchstone evaluates the resulting layouts as a unified benchmark. Extensive experiments demonstrate that LabBuilder significantly outperforms existing state-of-the-art methods, producing laboratory environments that are realistic and valid under modeled geometric, chemical-safety, and navigation constraints.

## 1. Introduction

Figure 1 illustrates three representative categories of problems and the corresponding solutions, core components

*Equal contribution [1]Shanghai Artificial Intelligence Laboratory [2]Wuhan University [3]Beihang University [4]Peking University [5]Tsinghua University [6]Shanghai Jiaotong University [7]The Chinese University of Hong Kong. Correspondence to: Dongzhan Zhou <zhoudongzhan@pjlab.org.cn>, Shixiang Tang <shixiang-tang@cuhk.edu.hk>.

*Proceedings of the 43rd International Conference on Machine Learning*, Seoul, South Korea. PMLR 306, 2026. Copyright 2026 by the author(s).

(Asset Knowledge Base and Chemical Knowledge Base), and the advantages of LabBuilder over existing methods. Automated laboratories (Tom et al., 2024; Collins et al., 2025; MacLeod et al., 2020) have the potential to accelerate scientific discovery by minimizing human intervention in repetitive experimental procedures. However, effective automation requires not only capable agents (M. Bran et al., 2024; Ghareeb et al., 2025) but also **safe and executable environments** in which laboratory protocols can be spatially supported. In practice, designing such environments directly in real laboratories could be costly and time-consuming, particularly at the level of scene layout, where small geometric or spatial changes can invalidate protocols or introduce hazards. Therefore, simulator-based 3D laboratories provide a scalable and cost-efficient platform for developing and evaluating laboratory scene layouts under modeled safety and reachability constraints, enabled by recent advances in interactive simulation (Mittal et al., 2025; Savva et al., 2019) and 3D scene generation (Yang et al., 2024b).

Despite impressive progress, most 3D scene generation methods target household indoor environments, emphasizing visual plausibility and geometric consistency while loosely constraining functionality, which makes them unsuitable for laboratory scenarios. This visual–functional gap arises because inputs are typically generic household-style descriptions that model objects as visually distinct assets with limited functional semantics, and thus cannot encode complex experimental protocols (Deitke et al., 2020), such as equipment-specific operational constraints or chemical properties of reagents. Moreover, existing methods mainly enforce static geometric validity and visual plausibility during generation (Li et al., 2023), leaving protocol grounding and agent reachability to post hoc evaluation. In laboratories, however, these properties are fundamental and should be considered at design time to support safe navigation and multi-step experimental workflows.

In this work, we propose LabBuilder, a protocol-grounded framework that compiles free-form experimental requests into asset-grounded, safety-aware laboratory layouts. To process the multi-facet protocols, LabForge acts as an in-

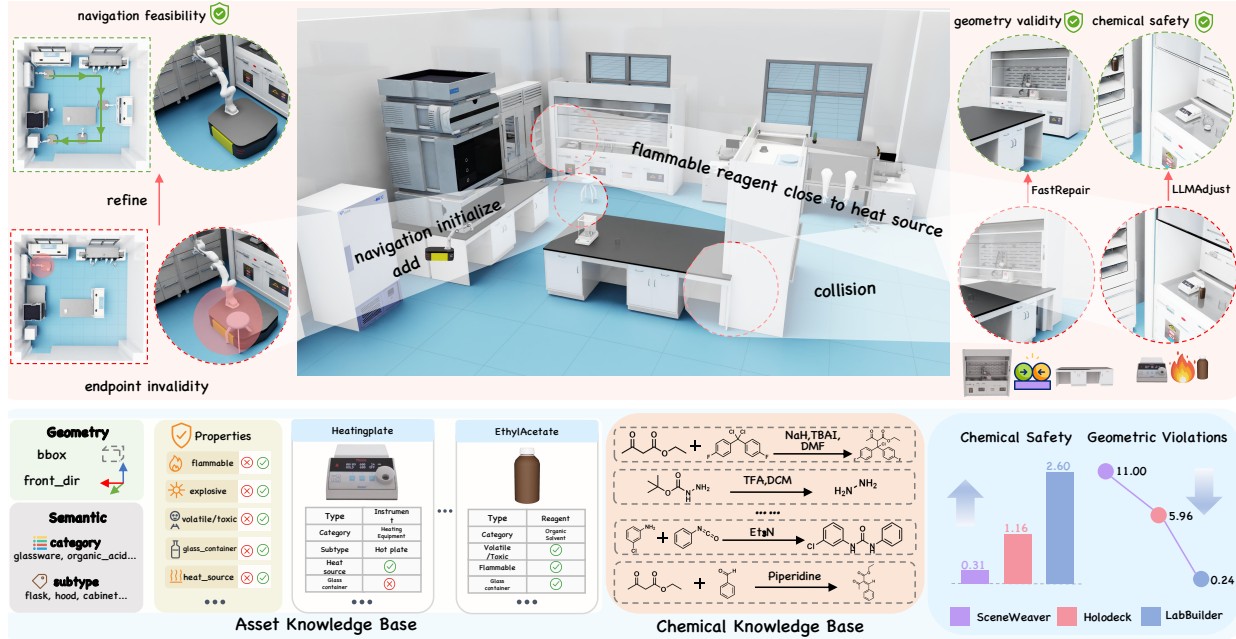

*Figure 1.* **Overview of LabBuilder.** We construct both an asset knowledge base and a chemical knowledge base from heterogeneous inputs, and synthesize asset-grounded experimental protocols. Based on these protocols, we generate laboratory layouts that ensure navigational feasibility, geometric validity, and chemical safety. LabBuilder achieves the best performance compared to existing methods.

terface that compiles free-form descriptions into structured protocols. This process is supported by two curated knowledge bases: an asset knowledge base containing functional and chemical semantics and a chemistry knowledge base distilled from real-world experimental datasets (Liu et al., 2024; Zhang et al., 2025), ensuring a high-level yet precise specification of the experimental workflow. Conditioned on the protocols, LabBuilder employs a hierarchical approach that factorizes scene synthesis into room-level zoning and desktop-level organization. To improve the generated layouts under geometric and chemical-safety constraints, we implement a geometric and chemical optimization strategy guided by a multi-objective reward function, which accounts for geometric validity and protocol-specific chemical safety. Furthermore, LabBuilder incorporates a navigation-aware refinement strategy to guarantee the reachability for autonomous robotic agents.

To systemically evaluate the value of generated laboratory scenes, we propose LabTouchstone, a comprehensive evaluation suite for both scene layout generation and agent navigation. For layout assessment, our framework quantifies performance across four key dimensions: geometric compliance, feasibility, chemical safety, and semantic plausibility. Beyond static generation, our suite supports protocol-derived point-goal navigation assessment (Cai et al., 2025), where experiments reveal that laboratory environments also pose unique challenges for existing navigation policies.

Our contributions are summarized as follows.

(i) We introduce a scalable meta-laboratory dataset consisting of an asset knowledge base and a chemistry knowledge base; the LabForge automated annotation engine enables continual expansion by seamlessly incorporating new assets and experiments. (ii) The LabGen framework can produce realistic, protocol-grounded laboratory layouts under geometric, chemical-safety, and reachability constraints. Extensive experiments demonstrate that LabGen surpasses existing methods by a notable margin. (iii) We establish LabTouchstone, a comprehensive metric suite for evaluating generated laboratory scenes in layout and navigation across multiple complementary dimensions.

## 2. Related Work

**Learning-Based Indoor Scene Generation.** Indoor virtual environments are traditionally built via manual authoring or rule-based procedural design (Kolve et al., 2017; Deitke et al., 2022), offering controllability but limited scalability. 3D scanning and reconstruction improve visual fidelity (Xia et al., 2018; Ramakrishnan et al., 2021), yet often leave interaction semantics and functional properties implicit, limiting task-driven evaluation (Savva et al., 2019). With large-scale 3D datasets, learning-based generators predict object categories and poses (Paschalidou et al., 2021; Wang et al., 2021; Tang et al., 2024) and produce plausible layouts on benchmarks such as 3D-FRONT (Fu et al., 2021). However, these models inherit closed-set vocabularies and taxonomies biased toward generic indoor domains (Paschalidou et al., 2021; Wang et al., 2021). More broadly, prior

paradigms emphasize visual plausibility and geometric validity (Wang et al., 2021; Tang et al., 2024), which is insufficient for chemical laboratories where workflows and safety require explicit, machine-checkable semantics (Yang et al., 2024a).

**Language-Conditioned Scene Synthesis with Constraints.** LLMs and multimodal foundation models enable language-conditioned, open-vocabulary scene synthesis (Fu et al., 2024). Many systems use LLM planners to translate language into structured scene decisions (Feng et al., 2023; Lin et al., 2023b) and combine them with text-to-3D pipelines for objects and appearances (Poole et al., 2022; Lin et al., 2023a; Höllein et al., 2023). However, directly emitting numerical layouts can be physically inconsistent (Lin et al., 2023b), motivating constraint enforcement via structured intermediates, post-processing, or external geometric/physical modules (Sun et al., 2025; Huang et al., 2025), as well as multi-agent generation for improved consistency (Çelen et al., 2024). Even so, downstream task requirements are often implicit (Sun et al., 2025): collision-free layouts may remain unusable due to unreachable operational poses, navigation dead ends, or domain-specific safety violations (Li et al., 2023; Szot et al., 2021; Yang et al., 2024a). This motivates workflow grounding for laboratory environments, where protocol consistency and safety compliance are central.

**Laboratory Automation and Protocol Representation.** Laboratory automation and chemical protocol representation have been studied through XDL/Chemputer, its $\chi$DL logical-control-flow extension, and AutoDSL (Mehr et al., 2020; Šiaučiulis et al., 2024; Shi et al., 2024), which focus on digitizing, structuring, or executing chemical procedures. LabBuilder addresses a complementary layer: grounding protocol semantics into 3D laboratory layouts under geometric, chemical-safety, and reachability constraints.

**Executable Environment Generation for Embodied Agents.** Environment generation for embodied intelligence targets task diversity and generalization (Deitke et al., 2022; Li et al., 2023), typically evaluated with episodic success, path length, or visual fidelity (Kolve et al., 2017). Such benchmarks often focus on short-horizon interactions (Deitke et al., 2020), whereas chemical laboratories require long-horizon protocol grounding under safety compliance and agent-centric feasibility (Li et al., 2023). LabBuilder follows this workflow-centric view by grounding laboratory semantics and protocol structure, then improving layouts via verification-driven, constraint-aware optimization and targeted repairs (Yang et al., 2024b; Sun et al., 2025). Accordingly, closing the visual–functional gap calls for protocol grounding in the generation loop rather than post hoc evaluation.

*Table 1.* Distribution of per-protocol over 30 experiments.

| Count type | Mean | Min | Max | Std |
|---|---|---|---|---|
| Reagents | 5.27 | 3 | 9 | 1.86 |
| Instruments | 9.87 | 7 | 14 | 1.81 |
| Steps | 9.00 | 6 | 11 | 1.46 |
| Moves | 4.30 | 3 | 6 | 1.15 |

## 3. LabBuilder

### 3.1. LabForge

LabForge compiles heterogeneous experimental inputs into a unified representation via a two-stage pipeline, as illustrated in Figure 2. Specifically, heterogeneous assets are first organized into a structured asset knowledge base $\mathcal{A}$, and asset-grounded experimental protocols $\mathcal{P}$ are then synthesized and checked from free-form experimental descriptions with reference to $\mathcal{A}$. This unified formulation provides a consistent foundation for protocol grounding, safety verification, and downstream laboratory layout generation.

**Asset Annotation.** We organize all laboratory assets into a unified annotation knowledge base that supports layout synthesis, protocol grounding, and safety verification. Each asset is annotated along three complementary dimensions: geometry, semantics, and domain/safety attributes. We construct a comprehensive asset annotation set that covers 176 laboratory entities, including reagents, experimental instruments, and room-level infrastructure (see Appendix A for details). Each asset is consistently annotated with attributes discussed above, which form a semantically rich and safety-aware foundation that supports protocol grounding and layout synthesis.

**Asset-Grounded Protocol Synthesis with Verification.** Given a free-form experimental description $x$, the protocol synthesis module generates a structured protocol grounded to the asset knowledge base $\mathcal{A}$, as illustrated in Figure 2. We first construct an experiment library from professional chemical literature and curated chemistry databases (Liu et al., 2024; Zhang et al., 2025). Each entry is represented by a chemical equation that captures the underlying reaction semantics and a sequence of atomic actions that specifies the operational steps required for accomplishing the experiment. The experiment library covers seven representative reaction types, including substitution, protection/deprotection, condensation, cyclization, redox, functional group transformation, and alkylation/acylation, reflecting a broad and realistic range of chemical procedures. We then retrieve relevant experiments or fragments from the experiment library using semantic and keyword matching to form a context $\mathcal{C}$. Conditioned on $(x, \mathcal{C})$, an LLM synthesizes an asset-grounded protocol $\mathcal{P}$, where asset mentions are normalized to entries in $\mathcal{A}$ and the protocol is checked for schema cor-

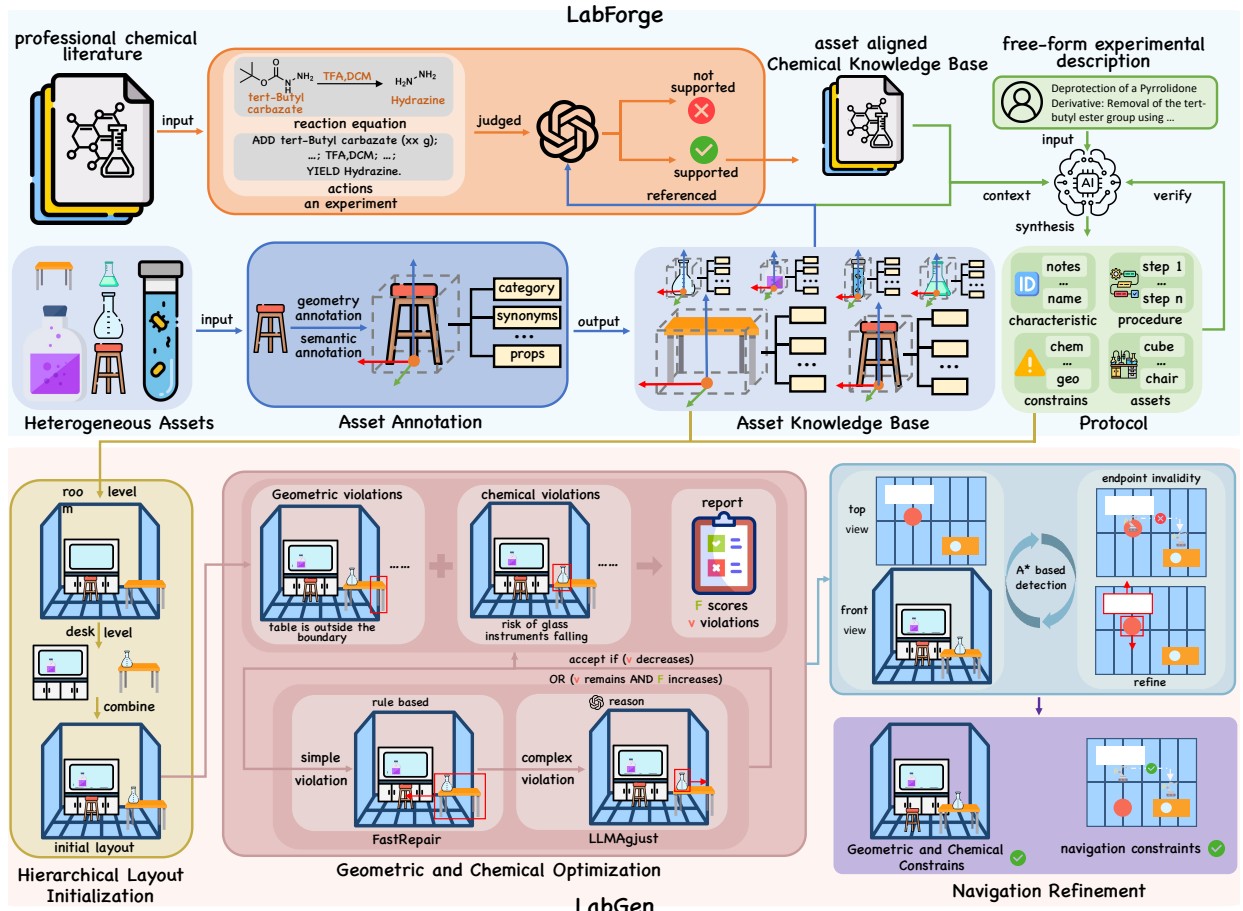

*Figure 2.* **LabBuilder framework combining LabForge and LabGen.** The system constructs an asset knowledge base from heterogeneous assets, synthesizes asset-grounded protocols from free-form experimental descriptions, and generates protocol-grounded laboratory layouts via hierarchical initialization, Geometric and Chemical Optimization, and navigation refinement.

rectness. Finally, we apply constraint-based checks against the asset knowledge base, including asset existence, valid step locations, and supported safety constraints.

As summarized in Table 1, across 30 curated experiments, each protocol contains on average 5.27 reagents, 9.87 instruments, 9.00 steps, and 4.30 navigation actions. We observe substantial variability across protocols (e.g., 3–9 reagents and 7–14 instruments), while common solvents and reagents (e.g., ethyl acetate and dichloromethane (DCM)) and standard glassware (e.g., round-bottom flasks and beakers) are frequently reused across reaction categories, consistent with typical laboratory practice. Overall, these statistics highlight both asset diversity and multi-step operational complexity, making the benchmark suitable for evaluating laboratory layout generation and navigation-aware feasibility.

### 3.2. LabGen

In this section, we present LabBuilder (Algorithm 1), a hierarchical synthesis framework for generating laboratory layouts that are physically feasible, chemically safe, and kinematically accessible for robotic agents.

**Hierarchical Layout Initialization.** Generating a complete laboratory scene in a single LLM pass (Team et al., 2025) is often intractable due to the vast configuration space of heterogeneous assets. We therefore factorize layout generation into two steps: (i) room-level zoning, which identifies the functional regions required by $\mathcal{P}$ and allocates them spatially within the room; and (ii) desktop-level organization, which populates each surface $s$ with protocol-relevant assets under boundary and non-overlap constraints. Given an experimental description $x$, a synthesized protocol $\mathcal{P}$, and an asset knowledge base $\mathcal{A}$, the process is modeled as:

$$\begin{aligned} \text{Room-Level:} \quad & (\mathcal{R}, \pi) \sim p_\theta(\mathcal{R}, \pi \mid x, \mathcal{P}, \mathcal{A}), \\ \text{Desktop-Level:} \quad & \mathcal{D}_s \sim p_\theta(\mathcal{D}_s \mid x, \mathcal{P}, \mathcal{A}, s, \mathcal{R}, \pi), \end{aligned} \quad (1)$$

where $\mathcal{R}$ denotes the set of room-level assets (*e.g.*, workbenches, fume hoods) with 6-DoF poses, $\pi$ represents the assignment variable, and $\mathcal{D}_s$ denotes the desktop-level layout for each surface $s$. Merging these hierarchical placements yields the initial candidate layout $\mathcal{L}_0$.

---

**Algorithm 1** LabBuilder

---

**Input:** Experimental description $x$, Protocol $\mathcal{P}$, asset knowledge base $\mathcal{A}$

**Output:** Optimal layout $\mathcal{L}^\star$

 1: // *Phase 1: Hierarchical Layout Initialization*
 2: Sample room-level zoning $(\mathcal{R}, \pi)$ using Eq. 1
 3: Sample desktop-level organization $\mathcal{D}$ using Eq. 1
 4: Initialize layout $\mathcal{L}_0 \leftarrow \mathcal{R} \cup \mathcal{D}$
 5: // *Phase 2: Geometric and Chemical Optimization*
 6: **for** level $\in \{\text{Room}, \text{Desktop}\}$ **do**
 7:    **repeat**
 8:       $I_t \leftarrow \psi(\mathcal{L}_t)$
 9:       $\mathcal{L}_{\text{prop}} \leftarrow \Phi(\mathcal{L}_t, \mathcal{P}, \mathcal{A})$
10:       Calculate violations $v_{\text{prop}} \leftarrow v(\mathcal{L}_{\text{prop}}, \mathcal{P}, \mathcal{A})$
11:       **if** $v_{\text{prop}} = v(\mathcal{L}_t, \cdot)$ **and** $\mathbb{F}(\mathcal{L}_{\text{prop}}, \cdot) > \mathbb{F}(\mathcal{L}_t, \cdot)$ **or** $v_{\text{prop}} < v(\mathcal{L}_t, \cdot)$ **then**
12:          $\mathcal{L}_t \leftarrow \mathcal{L}_{\text{prop}}$
13:       **end if**
14:    **until** converged **or** max iterations reached
15: **end for**
16: // *Phase 3: Navigation-Aware Refinement*
17: **while** $f_{\text{reach}}(\mathcal{L}_t, \mathcal{P}, \mathcal{A}) = 0$ **do**
18:    $\mathcal{L}_t \leftarrow \Upsilon(\mathcal{L}_t, \mathcal{P}, \mathcal{A})$
19: **end while**
20: $\mathcal{L}^\star \leftarrow \mathcal{L}_t$

---

**Geometric and Chemical Optimization.** To improve the layout, we formulate layout generation as a constrained optimization problem. The goal is to obtain a layout $\mathcal{L}^\star$ guided by a multi-objective reward function $\mathbb{F}$, representing the alignment between the synthesized spatial configuration and the task requirements:

$$\mathcal{L}^\star = \mathcal{L}_T, \quad \mathcal{L}_{t+1} = \Phi(\mathcal{L}_t, \mathcal{P}, \mathcal{A}; \mathbb{F}), \qquad (2)$$

where $\mathcal{L}_0$ is the initial layout and $T$ denotes the final repair iteration. The function $\mathbb{F}(\mathcal{L}, \mathcal{P}, \mathcal{A})$ is defined as a weighted combination of three complementary dimensions, where $w_{\text{geo}}$ and $w_{\text{chem}}$ are hyper-parameters:

$$\mathbb{F}(\mathcal{L}, \mathcal{P}, \mathcal{A}) = w_{\text{geo}} \, f_{\text{geo}}(\mathcal{L}, \mathcal{A}) + w_{\text{chem}} \, f_{\text{chem}}(\mathcal{L}, \mathcal{P}, \mathcal{A}). \quad (3)$$

Here, $f_{\text{geo}}$ measures geometric validity from boundary and collision constraints encoded in $\mathcal{A}$, and $f_{\text{chem}}$ measures protocol-grounded chemical safety by aggregating satisfaction ratios of constraints derived from hazard annotations in $\mathcal{A}$ and cues in $\mathcal{P}$. To solve this, we employ a Geometric and Chemical Optimization strategy guided by the violation-prioritized policy. Let $v(\mathcal{L}, \mathcal{P}, \mathcal{A})$ count the hard-constraint violations (*e.g.*, geometric intersections or critical safety hazards). The transition from $\mathcal{L}_t$ to $\mathcal{L}_{t+1}$ is accepted if $v(\mathcal{L}_{t+1}, \cdot) < v(\mathcal{L}_t, \cdot)$, or if $v(\mathcal{L}_{t+1}, \cdot) = v(\mathcal{L}_t, \cdot)$ and $\mathbb{F}(\mathcal{L}_{t+1}, \cdot) > \mathbb{F}(\mathcal{L}_t, \cdot)$.

Specifically, the hybrid operator $\Phi$ integrates two mechanisms: *FastRepair*, which resolves boundary and collision conflicts; *LLMAdjust*, which leverages the LLM (Team et al., 2025) to propose pose adjustments for safety and workflow constraints, followed by the same verification before acceptance. To manage the high-dimensional search space, the optimization process $\Phi$ is executed in two levels. It begins at the room-level, focusing on the large equipment and functional zones. When violations no longer decrease and $\mathbb{F}$ plateaus, the process shifts to the desktop-level. This subsequent stage adjusts the placement of smaller instruments and reagents, ensuring precise alignment with the experimental protocol until final convergence.

**Navigation-Aware Refinement.** A geometrically valid layout may still be unusable if the robotic agent cannot access the required instruments. To bridge the gap between static layout synthesis and protocol-derived reachability, we introduce a Navigation-Aware Refinement strategy. We first project the 3D scene into a 2D occupancy grid, applying obstacle dilation to account for the agent's collision volume. The $A^*$ algorithm (Hart et al., 1968) is then employed to determine the feasibility of the path, ensuring a shortest, collision-free route. For each protocol step, start and goal positions are extracted from the protocol $\mathcal{P}$, grounded to instantiated assets, and projected onto reachable free-space regions. We identify three navigation failures: (i) Endpoint invalidity, where the start or goal position is occupied by obstacles; (ii) Boundary violation, where the position is outside the valid room space; (iii) Path unreachability, where endpoints are topologically disconnected due to obstacles. These failures are summarized by a binary navigation feasibility:

$$f_{\text{reach}}(\mathcal{L}, \mathcal{P}, \mathcal{A}) = \begin{cases} 1, & \text{if no failure occurs} \\ 0, & \text{if any failure occurs} \end{cases} \qquad (4)$$

If $f_{\text{reach}} = 0$, we iteratively invoke a refinement operator $\Upsilon$ to the layout:

$$\mathcal{L}_{t+1} = \Upsilon(\mathcal{L}_t, \mathcal{P}, \mathcal{A}). \qquad (5)$$

This process, detailed in Appendix C, iteratively improves the layout until all navigation constraints are satisfied or the maximum iteration limit is reached. This process improves the final layout $\mathcal{L}^\star$ with respect to protocol-derived navigation requirements.

### 3.3. LabTouchstone

To enable systematic evaluation of protocol-grounded laboratory layout generation. we propose LabTouchstone, a suite of metrics that quantifies layout quality from four complementary perspectives: *geometric compliance*, *feasibility success rate*, *chemical safety*, and *semantic plausibility*. Together, these criteria capture whether a generated lab is geometrically valid, safety-aware, semantically reasonable, and practically usable for automated experimentation.

**Geometric Compliance.** Following the evaluation protocol of SceneWeaver (Yang et al., 2025), we assess generated scenes using three metrics: (i) asset count (*Obj*), the average number of objects per scene; (ii) object–object collisions (*CN*); and (iii) boundary validity (*OB*), with the latter two reported as violation counts to indicate Geometric Compliance.

**Feasibility Success Rate.** Feasibility success rate (FSR) evaluates whether a generated layout is feasible for a target protocol under modeled asset and navigation constraints. A layout is feasible only if it satisfies both asset availability and navigation feasibility, computed as independent ratios. Asset availability is a scenario-level binary indicator (1 if all required assets are present, 0 otherwise), and the final score is averaged over all generated scenarios. Navigation feasibility evaluates whether an agent can traverse protocol-derived step-to-step transitions, computed as the fraction of transitions for which a collision-free path can be planned with $A^*$ across scenarios; we also report protocol-level navigation, where a protocol succeeds only if all ordered transitions are feasible.

**Chemical Safety.** Chemical safety assesses adherence to critical safety protocols based on asset attributes and spatial relations. We focus on four key constraints: (i) isolation of flammable reagents from heat sources (*Flam.*); (ii) appropriate storage of reagents (*Store*); (iii) spatial separation of incompatible chemicals (*Incomp.*); and (iv) secure placement of glassware away from table edges (*Glass*). For distance-dependent constraints, we employ a *continuous satisfaction function*. Unlike binary pass/fail judgments, this function maps the ratio of actual-to-required distance to a scalar score, providing a granular assessment of risk. Furthermore, recognizing that safety is only relevant in actionable scenarios, we define an *asset-availability-weighted chemical score* to gate safety evaluation by asset availability:

$$\bar{s} = \frac{1}{N} \sum_{i=1}^{N} g_i \cdot s_i, \quad g_i \in \{0, 1\}, \tag{6}$$

where $s_i$ denotes the chemical safety score for scenario $i$, and $g_i$ is a binary indicator that equals 1 only if the scenario satisfies asset availability, and 0 otherwise.

**Semantic Plausibility.** We additionally introduce an LLM-based semantic evaluation (Hurst et al., 2024) on rendered layouts, covering three aspects: realism (*Real*), layout rationality (*Lay*), and completeness (*Comp*), with each aspect scored out of 10.

**Point-Goal Navigation.** We adopt point-goal navigation as the benchmark for evaluating navigation performance. Unlike prior benchmarks that use sparse scenes with few obstacles, our environments consist of complex chemical laboratories that more closely reflect real-world conditions.

An episode is considered successful if the agent reaches the target within a 0.5 m threshold, consistent with the precision required for protocol-derived navigation in cluttered laboratory scenes. We report Success Rate (SR) and Success weighted by Path Length (SPL) as standard metrics to measure task completion and path efficiency relative to the shortest path. For each scene, identical start–goal pairs are used to evaluate transitions between workbenches in protocol-derived workflows.

## 4. Experiments

In this section, we conduct a comprehensive evaluation of **LabBuilder**. We compare it against indoor scene generation baselines, analyze the benefit of evaluation-feedback-guided iterative optimization, and perform ablation studies on key components. We also report qualitative visualizations and protocol-derived navigation results.

**Scene Generation for Laboratory.** We reproduce Holodeck and SceneWeaver using the same 30 experiments, with natural-language descriptions as input, and evaluate them under the same metric suite. For fairness, all methods are evaluated using only our annotated dataset. We adapt Holodeck and SceneWeaver to construct scenes by retrieving assets from this dataset. We additionally evaluate Holodeck+, which provides Holodeck with the same structured protocol information as LabBuilder, including required assets and chemical safety constraints. Table 2 shows that our method generates protocol-grounded lab scenes, highlighting the benefit of integrating safety and reachability constraints into layout generation. For geometric compliance, our method achieves almost zero boundary and collision violations (OB=0.07, CN=0.17), indicating that the generated layouts are geometrically well-formed. In contrast, Holodeck suffers from severe boundary violations (OB=10.8), and SceneWeaver also exhibits geometric inconsistencies (OB=5.61, CN=0.35). Holodeck+ improves asset coverage and several safety metrics, such as Flam. $(0.239 \rightarrow 0.723)$ and Incomp. $(0.087 \rightarrow 0.613)$, but worsens geometric violations (OB $10.8 \rightarrow 15.2$) and storage/glass placement, suggesting that richer prompting alone cannot satisfy layout constraints jointly. Our method consistently satisfies protocol-level feasibility with the high asset availability (0.833) and navigation feasibility (0.966). Note that navigation feasibility cannot be evaluated for the baselines as they lack protocol modeling. The geometry-oriented layout generation in SceneWeaver lacks essential laboratory assets with asset availability 0.226, which demonstrates its limitation in protocol grounding and workflow support.

Our method also achieves competitive performance across all four safety constraints, indicating that hazardous interactions can be mitigated (e.g., the separation of flammable reagents) through sufficient asset arrangement. In contrast,

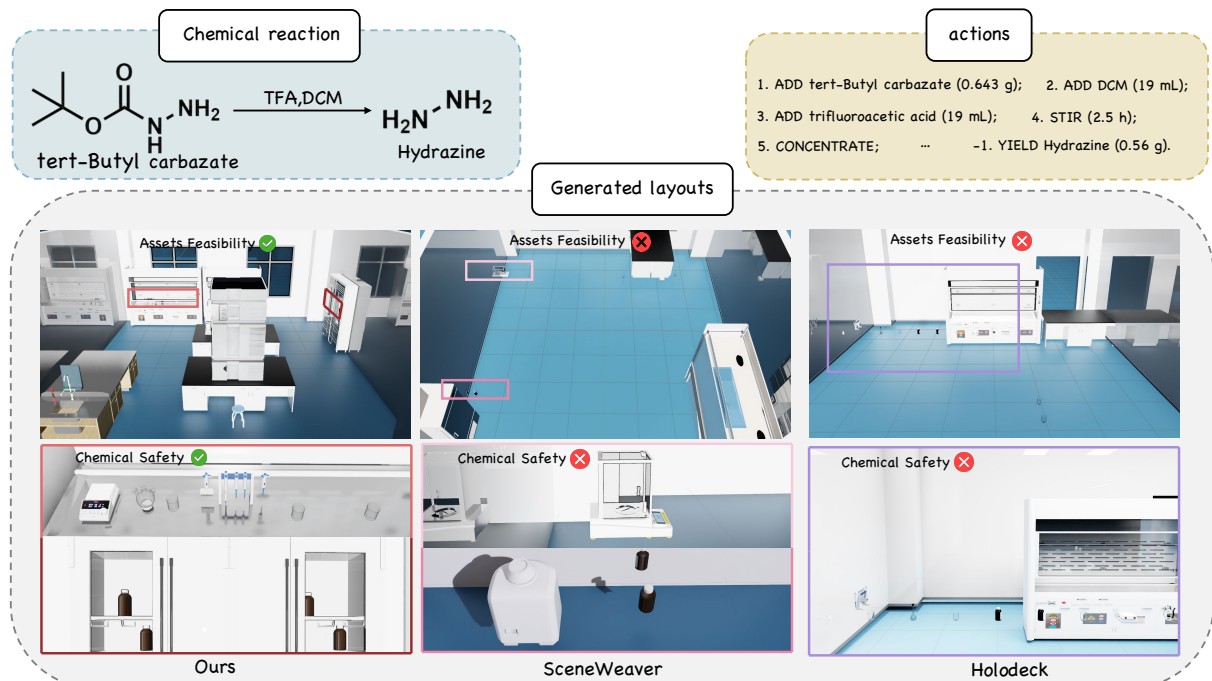

*Figure 3.* **Qualitative comparison** of layouts for the same chemical reaction by our method, SceneWeaver, and Holodeck. The top panel shows the target reaction and ordered protocol actions, while the bottom panels compare generated layouts under asset feasibility and chemical-safety checks. Green marks indicate satisfied checks, and red marks or highlighted boxes indicate representative failures.

*Table 2.* **Comparison with existing methods** on 30 experiments. We report geometry violations, feasibility success rate, chemical safety satisfaction, and LLM-based semantic scores. Holodeck+ uses the same structured protocol information as LabBuilder. (-) indicates not applicable due to missing protocol grounding.

| Method | Geometry | | | FSR | | Chemistry | | | | Visual & Semantics | | |
|---|---|---|---|---|---|---|---|---|---|---|---|---|
| | Obj | OB↓ | CN↓ | Asset↑ | Nav.↑ | Flam.↑ | Store↑ | Incomp.↑ | Glass↑ | Real↑ | Lay↑ | Comp↑ |
| Holodeck | 15.4 | 10.8 | 0.20 | 0.700 | – | 0.239 | 0.583 | 0.087 | 0.252 | 6.14 | 5.61 | 4.61 |
| SceneWeaver | 10.3 | 5.61 | 0.35 | 0.226 | – | 0.097 | 0.000 | 0.194 | 0.020 | 5.30 | 4.57 | 3.20 |
| Holodeck+ | 19.8 | 15.2 | 0.33 | 0.733 | – | 0.723 | 0.000 | 0.613 | 0.041 | 6.27 | 5.43 | 5.20 |
| Ours | 23.2 | 0.07 | 0.17 | 0.833 | 0.966 | 0.725 | 0.801 | 0.716 | 0.364 | 8.23 | 9.00 | 9.10 |

SceneWeaver (Yang et al., 2025) collapses on storage safety (Store=0.000), showing that reagents are frequently placed in fundamentally unsafe locations, while Holodeck remains weak on incompatibility separation (Incomp=0.087). Finally, our method also surpasses baselines in visual and semantic quality (Real=8.23, Comp=9.10), indicating that modeled safety and reachability constraints do not come at the expense of realism.

**Qualitative Comparison.** As illustrated in Figure 3, our method outperforms Holodeck (Yang et al., 2024b) and SceneWeaver (Yang et al., 2025) in both asset completeness and the precision of desktop-level placement. Our generated scenes are largely free of inter-object collisions and boundary violations, whereas baseline methods frequently suffer from large assets extending beyond scene boundaries.

Moreover, baseline results for small desktop-level objects often exhibit unrealistic artifacts, including hovering objects or items incorrectly placed on the floor. In addition, our layouts satisfy key chemical safety constraints (e.g., proper storage and separation), while the baselines exhibit unsafe configurations, as indicated in the figure.

**Laboratory Navigation.** Table 4 reports point-goal navigation performance of iPlanner (Yang et al., 2023) and NavDP (Cai et al., 2025) in the generated laboratory scenes, evaluated before and after Navigation-Aware Refinement. Both baselines show moderate performance, with iPlanner achieving 0.350 success and 0.332 SPL, and NavDP reaching 0.439 success and 0.367 SPL, suggesting that the laboratory scenes form a challenging out-of-distribution setting compared to household environments. Navigation

*Table 3.* **Ablation study** on asset annotations, protocol guidance, Geometric and Chemical Optimization (Geom. & Chem. Opt.), and Navigation-Aware Refinement (nav. opt.). Protocol-level navigation succeeds only if all ordered transitions in a protocol are feasible. (-) indicates not applicable.

| Method | Geometry | | | FSR | | | Chemistry | | | | Visual & Semantics | | |
|---|---|---|---|---|---|---|---|---|---|---|---|---|---|
| | Obj | OB↓ | CN↓ | Asset↑ | Step Nav.↑ | Prot. Nav.↑ | Flam.↑ | Store↑ | Incomp.↑ | Glass↑ | Real↑ | Lay↑ | Comp↑ |
| Ours (w/o annotation) | 23.3 | 0.25 | 0.36 | 0.786 | 0.952 | 0.857 | 0.452 | 0.583 | 0.701 | 0.355 | 7.14 | 6.54 | 5.54 |
| Ours (w/o protocol) | 28.3 | 2.80 | 3.30 | 0.300 | – | – | 0.098 | 0.000 | 0.179 | 0.161 | 5.03 | 5.28 | 3.14 |
| Ours (w/o Geom. & Chem. Opt.) | 23.2 | 2.73 | 1.43 | 0.833 | 0.959 | 0.767 | 0.182 | 0.000 | 0.367 | 0.128 | 7.20 | 7.00 | 7.63 |
| Ours (w/o nav. opt.) | 23.2 | 0.07 | 0.17 | 0.833 | 0.870 | 0.533 | 0.736 | 0.801 | 0.716 | 0.367 | 8.10 | 8.53 | 8.90 |
| Ours | 23.2 | 0.07 | 0.17 | 0.833 | 0.966 | 0.867 | 0.725 | 0.801 | 0.716 | 0.364 | 8.23 | 9.00 | 9.10 |

performance improves consistently for both methods after refinement. The benchmark clearly differentiates the two approaches, with NavDP consistently outperforming iPlanner on both metrics. This performance gap likely stems from differences in perceptual robustness and prior knowledge: iPlanner's monocular cues are brittle in densely structured layouts, whereas NavDP benefits from stronger learned traversability priors.

### 4.1. Ablation Study

We ablate four key components of LabBuilder: (i) asset annotations, (ii) protocol guidance, (iii) Navigation-Aware Refinement, and (iv) Geometric and Chemical Optimization. Unless otherwise specified, all variants use the same asset pool, protocol inputs, and evaluation metrics.

**Asset Annotations.** Disabling semantic/domain asset annotations (retaining geometry only) consistently degrades both *geometric compliance* and *chemical safety* (Table 3). Geometric violation rates increase substantially (e.g., OB $0.07 \rightarrow 0.25$, CN $0.17 \rightarrow 0.36$), indicating more collision- and boundary-related placement errors when category- and affordance-level cues are removed. More critically, asset-availability-weighted chemistry drops markedly (e.g., flammable isolation $0.725 \rightarrow 0.452$; proper storage $0.801 \rightarrow 0.583$), suggesting that hazard attributes and synonym grounding are necessary for reliably triggering the correct safety constraints. This also manifests in a sharp decline in semantic completeness (Comp $9.10 \rightarrow 5.54$), implying that geometry-only assets lead to layouts that are less aligned with laboratory functionality and workflow expectations.

**Protocol Guidance.** Removing structured protocol guidance yields the *largest* performance drop across metrics (Table 3). Feasibility collapses (asset availability falls from $0.833 \rightarrow 0.300$), indicating that required instruments and reagents are frequently missing or mismatched with the intended workflow. In addition, physical violations surge (e.g., OB $0.07 \rightarrow 2.80$, CN $0.17 \rightarrow 3.30$), rendering many scenes unusable. As a consequence, asset-availability-weighted chemistry becomes near-degenerate (e.g., Store $0.801 \rightarrow 0.000$, Flam $0.725 \rightarrow 0.098$), and semantic com-

*Table 4.* **Point-goal navigation** results in the generated scenes.

| Method | Success(↑) | SPL(↑) |
|---|---|---|
| iPlanner (w/o nav. opt.) | 0.350 | 0.332 |
| iPlanner | 0.425 | 0.412 |
| NavDP (w/o nav. opt.) | 0.439 | 0.367 |
| NavDP | 0.517 | 0.446 |

pleteness drops sharply (Comp $9.10 \rightarrow 3.14$). Together, these results confirm that structured protocols are essential for selecting the correct asset set and generating layouts under geometric, safety, and reachability constraints.

**Geometric and Chemical Optimization.** To isolate the effect of *layout* optimization, we disable Geometric and Chemical Optimization while keeping Navigation-Aware Refinement enabled (Table 3, w/o Geom. & Chem. Opt.). Without iterative refinement, geometric feasibility degrades markedly: boundary and collision violations increase from $0.07/0.17$ to $2.73/1.43$ (OB/CN), indicating that many layouts retain geometric violations even if navigation is optimized. This directly suppresses asset-availability-weighted chemical safety, with large drops in key constraints such as flammable isolation ($0.725 \rightarrow 0.182$) and incompatibility separation ($0.716 \rightarrow 0.367$). Semantic quality also declines, with completeness decreasing from 9.10 to 7.63. Overall, iterative Geometric and Chemical Optimization is crucial for repairing geometric errors and improving chemical-safety satisfaction. Consistently, enabling iterative refinement improves all safety constraints and semantic judgments, suggesting that iterative Geometric and Chemical Optimization yields more realistic organization and better workflow support.

**Navigation-Aware Refinement.** We validate the proposed Navigation-Aware Refinement from both the *scene generation* and *navigation policy* perspectives. First, Table 3 shows that disabling Navigation-Aware Refinement primarily affects reachability: step-level navigation drops from 0.966 to 0.870, and protocol-level navigation drops from 0.867 to 0.533 (w/o nav. opt.), while geometry and chemistry/semantic metrics remain largely comparable (e.g., OB/CN

0.07/0.17 vs. 0.07/0.17, and asset-availability-weighted chemistry and semantic scores change only marginally). This indicates that Navigation-Aware Refinement specifically targets connectivity failures rather than altering the overall asset composition or safety constraints. Table 4 further shows that navigation-aware refinement consistently improves performance across multiple planners.

## 5. Conclusion

We proposed LabBuilder, an end-to-end framework that grounds free-form experimental requests into structured protocols and generates laboratory layouts that are geometrically valid, chemically safer under evaluated layout-level constraints, and reachable for protocol-relevant navigation. LabBuilder integrates three core modules: LabForge for asset and protocol compilation, LabGen for hierarchical layout generation with geometric and chemical optimization and navigation-aware refinement, and LabTouchstone for systematic evaluation. Experimental results demonstrate that explicitly incorporating protocol grounding and safety constraints during generation yields substantially more layout-level feasible laboratory environments than existing methods. Despite these advancements, LabBuilder does not yet validate equipment specification-level requirements, such as heating ranges, pipette volumes, or fumehood ratings, nor does it verify physical manipulation feasibility for robotic execution. Extending LabBuilder toward specification-aware validation, manipulation-level execution checking, broader regulatory constraints, multi-protocol laboratory design, and sim-to-real transfer remains important future work.

## Impact Statement

This paper presents work whose goal is to advance the field of Machine Learning. There are many potential societal consequences of our work, none of which we feel must be specifically highlighted here.

## Acknowledgements

This work is supported by New Generation Artificial Intelligence-National Science and Technology Major Project(2025ZD0121802).

This work was supported by the JC STEM Lab of AI for Science and Engineering, funded by The Hong Kong Jockey Club Charities Trust, the MTR Research Funding (MRF) Scheme (CHU-24003), the Research Grants Council of Hong Kong (Project No. CUHK14213224).

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

# Appendix

## A. LabForge: Asset Annotation Details

For geometry, all assets are canonicalized into a shared coordinate system with a fixed up-axis and a consistent front direction, and their shapes are abstracted by an axis-aligned bounding box. This unified geometric representation enables consistent collision detection, boundary constraints, and physical comparability across heterogeneous asset categories. For semantics, each asset is associated with a structured semantic descriptor consisting of its category, subtype, bilingual synonyms, and a short description. This representation allows robust grounding from free-form protocol descriptions to canonical asset identities, aligning diverse surface forms to a single standardized entry and enabling reliable protocol-to-asset mapping. For domain and safety attributes, assets are annotated with a set of boolean chemical and hazard indicators, including flammability, explosiveness, volatility/toxicity, material type (e.g., glass container), and chemical roles such as acid, base, oxidizer, or reactive metal. These attributes directly instantiate chemical compatibility and safety constraints during layout generation and optimization.

### A.1. Geometric Preprocessing Details

Before annotation, we apply unified geometric preprocessing to all assets: (i) scale all asset dimensions to a unified metric unit; (ii) canonicalize the asset orientation to face the positive $y$-axis; (iii) set the placement reference point to the center of the asset's bottom surface. We use a Z-up coordinate convention throughout. This is an example of the assets annotated by us, which includes room-level assets and desktop-level assets.

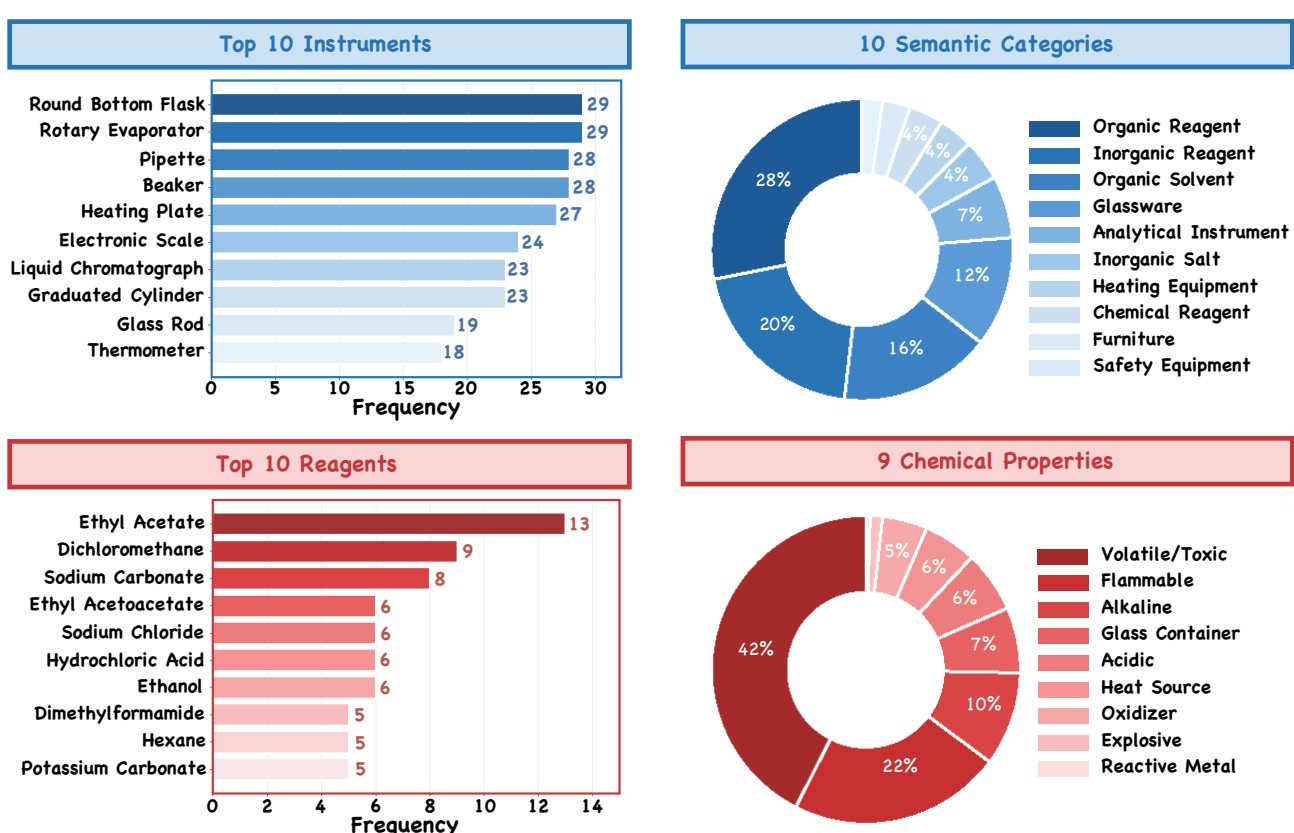

*Figure A.1.* LabForge data statistics of the asset knowledge base $A$ and the 30 synthesized experimental protocols.

### A.2. Phase 1: Physical Annotation Output Schema

The physical annotator takes raw asset data including name, type (instrument, reagent, or room facility), bounding box dimensions, and source pose/orientation information. Each asset is represented by an axis-aligned bounding box parameterized by *short side*, *long side*, and *height*, together with the canonical pose in the normalized coordinate system.

This output is used for spatial occupancy computation, collision detection, and boundary constraints.

**Asset(Bounding Box) Sample**

---

### *Case Asset Semantic & Geometry Specification - TestTube*

**Asset ID:** `TestTube`
**Asset Type:** Instrument

**[Semantic Description]**
**Category:** Glassware
**Subtype:** Test Tube
**Synonyms:**

- Culture tube

- Sample tube

**Description:**
A common piece of laboratory glassware consisting of a finger-like length of glass or clear plastic tubing, open at the top and closed at the bottom. It is typically used to hold, mix, or heat small quantities of chemicals.

**[Physical and Chemical Properties]**

- Flammable: **False**

- Explosive: **False**

- Volatile or Toxic: **False**

- Glass Container: **True**

- Heat Source: **False**

- Acid: **False**

- Base: **False**

- Oxidizer: **False**

- Active Metal: **False**

**[Spatial and Scene Information]**

**[Geometry Specification]**
**Bounding Box Dimensions (meters):**

$$\text{Short} = 9.996 \times 10^{-4}$$
$$\text{Long} = 2.414 \times 10^{-3}$$
$$\text{Height} = 2.351 \times 10^{-3}$$

**Scale Factor:** 1.0
**Front Direction:** (0, 1, 0)
**Coordinate System:** `z_up`

**[Asset Resource]**
**USD File Path:** `/assets/TestTube.usd`

---

**Notes:**
Standard laboratory glassware used for holding, mixing, or heating small amounts of chemicals. The asset is non-reactive and non-hazardous, making it suitable for general-purpose laboratory simulation and manipulation tasks.

### A.3. Phase 2: Semantic Annotation Prompting and Fields

We use a multimodal large language model (Google Gemini 3.0 Pro) as the semantic annotator, feeding both asset images and text metadata. The prompt explicitly defines asset types (instrument/reagent/facility) and includes in-context examples (e.g., beakers, ethanol, fume hoods). The model returns five fields:

(1) normalized name;

(2) fine-grained category (e.g., `glassware`, `organic_solvent`);

(3) subtype (e.g., `flask`, `alcohol`);

(4) bilingual synonym list (Chinese and English);

(5) a 1–2 sentence functional description.

---

**Semantic Annotation Prompt for Laboratory Assets**

You are a laboratory asset semantic annotation expert.
Your task is to output structured semantic information based on the provided asset information, and image(s) if available.

**Asset Type Definitions:**
{type_definitions}

**Image Instruction (if images are provided):**
Please carefully observe the provided image(s) and make judgments based on both the asset name and type.

**Asset Information:**

- Name: {name}

- Type: {asset_type}

**Required Output Format (JSON Fields):**

- canonical_name: Standard camelCase naming (e.g., ErlenmeyerFlask, HeatingPlate), aligned with existing naming conventions

- category: Fine-grained category (e.g., glassware, organic_solvent, safety_equipment)

- subtype: More specific subtype (e.g., flask, beaker, alcohol, acid)

- synonyms: List of synonyms (including both Chinese and English)

- description: Concise description (1–2 sentences)

**Examples:**
**Example 1:**
Input: name = "Beaker", type = "instrument", image shows a cylindrical glass container
Output:

    { "canonical_name": "Beaker",
    "category": "glassware",
    "subtype": "beaker",
    "synonyms": ["beaker flask"],
    "description": "A cylindrical glass container with a flat bottom and a spout, used for mixing and heating liquids." }

**Example 2:**
Input: name = "Ethanol", type = "reagent", image shows a reagent bottle
Output:

---

```
        { "canonical_name": "Ethanol",
        "category": "organic_solvent",
        "subtype": "alcohol",
        "synonyms": ["ethyl alcohol", "C2H5OH"],
        "description": "A colorless, volatile, flammable liquid alcohol, commonly used as a solvent and disinfectant."
        }
```

**Example 3:**
Input: name = "FumeHood", type = "room_asset", image shows a fume hood
Output:

```
        { "canonical_name": "FumeHood",
        "category": "safety_equipment",
        "subtype": "ventilation",
        "synonyms": ["fume cupboard", "safety hood"],
        "description": "A ventilation device designed to limit exposure to hazardous or toxic fumes, vapors, or dusts."
        }
```

**CRITICAL REQUIREMENTS:**

1. Output ONLY valid JSON, nothing else

2. Do NOT include markdown code blocks

3. Do NOT add explanatory text before or after the JSON

4. Ensure all strings are properly escaped and quoted

5. Ensure the JSON is complete and valid

6. Start the response with { and end with }

**Final Instruction:**
Output the JSON now.

## A.4. Phase 3: Domain Annotation for Laboratory Safety

We further infer lab-specific safety and experiment attributes (also via Gemini 3.0 Pro) to support safety constraints. The domain annotator outputs the following boolean attributes: flammability, explosiveness, volatility/toxicity, glass container, heat source, acid, base (alkali), oxidizer, and active metal. These attributes are subsequently compiled into our chemical safety constraint modeling and spatial partition strategy.

## A.5. Phase 4: Caption Annotation

After physical, semantic, and domain annotations, we generate a concise caption for each asset that summarizes typical usage scenarios, safety precautions, and recommended placement positions. We use the caption mainly for fast asset retrieval and robust matching in the protocol generation stage.

## B. LabForge: Chemistry Knowledge Base Curation and Protocol Generation Details

We construct the Chemistry Knowledge Base (i.e., the experiment library used for retrieval) from CHEMTRANS and OPENEXP by filtering experiments that are feasible under the current asset library. Feasibility is determined by whether required operations and instruments can be instantiated by assets in our Asset Knowledge Base (see Figure A.1).

## B.1. Retrieval Outputs and Content Scope

For an input description $x$, we use it as a query $q$ to perform hybrid retrieval with semantic similarity and keyword matching, and return the top-5 protocol fragments. Each retrieved fragment is associated with similarity scores and source identifiers. Retrieved content typically covers: (i) operation sequences, (ii) reagent/solvent choices, (iii) key experimental parameters (dosage, temperature, duration, stirring/monitoring), and (iv) potential risk points. These fragments are concatenated and organized into a Retrieval-Augmented Generation context $C$.

## B.2. Priors Used in Generation

We fuse $C$ with two types of priors:

- **Asset Knowledge Base Abstract.** This abstract constrains entity mentions to normalized asset names and valid category ranges, preventing out-of-inventory or inconsistent naming.

- **Rule-based Constraint Priors.** We enforce protocol structure and safety logic, e.g., requiring each step to include an explicit operation location, ensuring location selection adheres to safety principles, and generating corresponding chemical and spatial constraints.

## B.3. Generation Requirements

The generated protocol must satisfy:

1. **Completeness:** covering the full lifecycle including preparation, reaction, post-processing, purification, analysis, and cleaning.

2. **Detail:** specifying key parameters such as dosages, temperatures, times, and stirring/monitoring methods depending on experiment types.

3. **Constraints:** synthesizing actionable chemical and spatial constraints grounded in retrieval context and rule priors.

4. **Consistency:** using asset names consistent with the normalized Asset Knowledge Base.

## B.4. Automatic Validation Suite

After generation, we automatically validate: (i) all asset entities in the protocol are resolvable in the Asset Knowledge Base; (ii) every step contains required fields and is schema-complete; (iii) operation locations are legal and safety-consistent; (iv) all referenced objects and spatial regions in constraints are valid. We additionally verify hard constraints such as requiring reagent preparation as the first step and aligning hazardous operations with reasonable locations. Only protocols passing validation are forwarded to downstream layout generation, evaluation, and iterative optimization. One example protocol is shown below:

---

### *Case Protocol Semantic & Procedural Specification - Deprotection of tert-Butyl Ester*

**Protocol ID:** `exp_003`
**Protocol Type:** Chemical Reaction Protocol

**[Semantic Description]**
**Reaction Name:** Deprotection of tert-Butyl Ester using Trifluoroacetic Acid
**Reaction Category:** Protecting Group Removal
**Description:**
This protocol describes the acid-mediated deprotection of a tert-butyl ester functional group using trifluoroacetic acid (TFA) in dichloromethane (DCM). The reaction proceeds under mild conditions, followed by neutralization and liquid–liquid extraction to isolate the deprotected product.

**[Required Assets]**
**Reagents:**

---

- TertButylCarbazate — substrate containing acid-labile tert-butyl group

- Trifluoroacetic Acid (TFA) — acidic deprotection reagent

- Dichloromethane (DCM) — reaction solvent

- Sodium Carbonate — neutralization reagent

- Ethyl Acetate — extraction solvent

**Instruments:**

- Round Bottom Flask

- Beaker

- Separatory Funnel

- Graduated Cylinder

- Pipette

- Glass Rod

- Electronic Scale

- Rotary Evaporator

- Liquid Chromatograph

**[Physical and Chemical Constraints]**

- Corrosive Acid Involved: **True**

- Toxic Volatile Solvent: **True**

- Gas Evolution During Quench: **True**

- Flammable Organic Solvent: **True**

- High Temperature Required: **False**

- Pressurized System: **False**

**[Procedural Specification]**

1. Weigh the substrate (TertButylCarbazate) and transfer it into a round bottom flask.

2. Dissolve the substrate in dichloromethane under a fume hood.

3. Slowly add trifluoroacetic acid with stirring to initiate deprotection.

4. Allow the reaction to proceed at room temperature for approximately 2 hours.

5. Quench excess acid by slow addition of saturated sodium carbonate solution.

6. Transfer the mixture to a separatory funnel and extract with ethyl acetate.

7. Collect the organic layer and remove solvents using a rotary evaporator.

8. Analyze the crude product using liquid chromatography.

**[Spatial and Scene Information]**

- Primary Operation Location: `FumeHood`

- Extraction Location: `FumeHood`

- Solvent Removal Location: `RotaryEvaporator Station`

- Analysis Location: `Validation Platform`

**[Protocol Output]**

- Deprotected product (tert-butyl group removed)

- Reaction completeness verified by liquid chromatography

---

**Notes:**
This protocol represents a standard acid-labile protecting group removal widely used in organic synthesis. Special care must be taken due to the corrosive nature of trifluoroacetic acid and the volatility and toxicity of dichloromethane. All operations involving these reagents must be conducted in a properly ventilated fume hood.

## C. LabGen: Layout Generation and Iterative Optimization Details

This appendix documents implementation details omitted from Alg. 1 for clarity.

### C.1. Core Operator Definitions

**Definition of $p_\theta$.** The generator $p_\theta$ denotes the LLM-based hierarchical layout proposal module used in the initialization stage. It generates layouts in two stages:

$$\text{Room-Level:} \quad (\mathcal{R}, \pi) \sim p_\theta(\mathcal{R}, \pi \mid x, \mathcal{P}, \mathcal{A}), \tag{7}$$

$$\text{Desktop-Level:} \quad \mathcal{D}_s \sim p_\theta(\mathcal{D}_s \mid x, \mathcal{P}, \mathcal{A}, s, \mathcal{R}, \pi). \tag{8}$$

Here, $\mathcal{R}$ denotes room-level assets with 6-DoF poses, $\pi$ denotes protocol-to-zone assignments, $x$ is the experiment description, $\mathcal{P}$ is the structured protocol, $\mathcal{A}$ is the asset knowledge base, and $\mathcal{D}_s$ is the desktop layout for surface $s$. At the room level, the LLM receives room dimensions, the asset catalog, and protocol requirements, then outputs a JSON containing positions, orientations, and assignments $\pi$. At the desktop level, for each surface $s$, the LLM outputs placements in the local coordinate frame. The functional regions in $\pi$ are abstract workflow regions that guide grounding to assets and surfaces, rather than direct furniture labels.

**Definition of $\Phi$.** The operator $\Phi$ denotes one repair step in the geometric and chemical optimization stage:

$$\mathcal{L}_{t+1} = \Phi(\mathcal{L}_t, \mathcal{P}, \mathcal{A}; \mathbb{F}). \tag{9}$$

It combines geometric repair and semantic adjustment:

$$\Phi(\mathcal{L}_t, \mathcal{P}, \mathcal{A}) = \text{LLMAdjust}(\text{FastRepair}(\mathcal{L}_t, \mathcal{P}, \mathcal{A}), \mathcal{P}, \mathcal{A}). \tag{10}$$

A proposed update is accepted if it reduces the hard-constraint violation count, or if it keeps the violation count unchanged while improving the layout score $\mathbb{F}$.

**Definition of FastRepair.** FastRepair is a deterministic rule-based geometric repair procedure for boundary and collision violations:

1. **Boundary repair:** for each out-of-bound object, translate it inward by the minimum penetration distance.

---

**Algorithm 2** Navigation-Aware Refinement Operator $\Upsilon$

---

**Input:** Layout $\mathcal{L}_t$, protocol $\mathcal{P}$, asset knowledge base $\mathcal{A}$
**Output:** Refined layout $\mathcal{L}_{t+1}$

1: Project the 3D scene to a 2D occupancy grid with agent dilation
2: Generate protocol-derived start-goal pairs
3: Run A\* search over all start-goal pairs
4: Classify navigation failures as endpoint invalidity, boundary violation, or path unreachability
5: Compute unreachable paths $U_t$, total paths $N_t$, and unreachable rate $r_t = (U_t/N_t) \times 100\%$
6: **if** $U_t = 0$ **then**
7:     Return $\mathcal{L}_t$
8: **end if**
9: Generate rotation fixes and translation updates from navigation failures
10: Deduplicate updates per object and direction by keeping the largest absolute translation
11: Apply updates with workspace synchronization
12: Invoke $\Phi$ when additional geometric or chemical repair is needed
13: Return $\mathcal{L}_{t+1}$

---

2. **Collision repair:** for each colliding pair, compute a separating displacement from their 2D rotation-aware footprints and move the objects apart with a small safety margin.

3. **Workspace synchronization:** when a work surface is moved, translate all desktop objects on that surface by the same displacement to preserve their relative configuration.

4. **Iteration:** repeat the above steps until no geometric violations remain or the maximum number of repair rounds is reached.

**Definition of LLMAdjust.** LLMAdjust handles layout violations that require semantic reasoning, such as chemical-safety relations, workflow proximity, or complex rearrangements:

1. **Encode:** construct a prompt containing the current layout after FastRepair, violation diagnostics, chemical safety requirements, and the current optimization level.

2. **Query LLM:** ask the LLM to output JSON commands such as `move(object, pos)`, `rotate(object, angle)`, and `swap(object1, object2)`.

3. **Validate and apply:** reject adjustments that introduce new hard-constraint violations, and apply the valid ones.

**Definition of $\Upsilon$.** The operator $\Upsilon$ denotes one navigation-aware layout update:

$$\mathcal{L}_{t+1} = \Upsilon(\mathcal{L}_t, \mathcal{P}, \mathcal{A}). \tag{11}$$

It projects the 3D scene into a 2D occupancy grid with agent-radius dilation, runs A\* over protocol-derived start-goal pairs, classifies navigation failures, and generates rotation or translation updates for the responsible room-level assets. The detailed target generation, reachability checks, translation updates, rotation fixes, and deduplication rules are provided in the following subsections.

**C.2. Scene Representation and Coordinate Conventions**

- **Output format.** The final layout is exported as a structured JSON scene description directly importable by Isaac Sim.

- **Coordinate system.** We follow Isaac Sim's convention (Z-up). We place the origin at the front-left corner of the room floor. Each room-level object is represented by global $(x, y, z)$.

- **Local-to-global mapping for desktops.** Desktop objects are first sampled in a workbench-local surface frame whose origin is the front-left corner of the tabletop, with $X$ along width and $Y$ along depth; $Z$ is fixed to the tabletop height. Desktop poses are then mapped into the global room frame using the workbench global pose (including rotation) and the computed global position of its front-left tabletop corner.

## C.3. Geometric Priors and Collision/Boundary Checks

- **Asset occupancy.** Each asset in $A$ provides a 3D bounding box (short side, long side, height). We use its 2D projection on the horizontal plane to test boundary constraints and mutual collisions.

- **Rotation-aware footprint.** Since rotation changes the projected footprint, boundary detection and collision judgment are performed on rotation-aware footprints rather than axis-aligned boxes.

- **Hard global bounds.** Placement ranges are clipped by the internal room boundary (optionally deducting wall thickness) as hard constraints for room-level objects.

## C.4. Navigation Optimization

This subsection presents the overall workflow and core mathematical formulas of the navigation iterative optimization process. We model layout refinement as an iterative procedure driven by navigation reachability feedback: in each iteration, we first perform navigation analysis and compute reachability metrics, then update the layout by applying the recommended rotations and translations, optionally invoke a layout optimizer to repair physical constraints, and finally re-run navigation analysis. The process repeats until all paths are reachable or the maximum number of iterations is reached.

**Overall Iterative Workflow**    Let $T$ denote the maximum number of iterations. For each iteration $t$ ($t = 1, \ldots, T$), the optimization proceeds as follows:

1. **Navigation reachability analysis**: Invoke the navigation analysis module to evaluate the reachability of all navigation paths in all scenes (stored in the data directory). The module outputs a reachability label (reachable / unreachable) for each path, together with local adjustment suggestions (including rotation fixes and position translations) for unreachable paths.

2. **Computation of convergence metrics**: Aggregate the analysis results over all scenes and all paths, and compute the number of unreachable paths $U_t$, the total number of paths $N_t$, and the unreachable rate $r_t$ (see formulas below).

3. **Convergence check**: If $U_t = 0$, i.e., all paths are reachable, the algorithm is considered converged and the iteration terminates early; otherwise, proceed to the next step.

4. **Layout update**:

   (a) *Rotation fixes*: For platforms or devices that are detected as facing a wall, apply the suggested rotation angle to the platform/device itself, and perform a rigid-body rotation for all objects placed on that platform, so that their relative geometric configuration is preserved.

   (b) *Position adjustments*: For the translation suggestions produced by the analysis, deduplicate them per object and per direction, and then translate the platform and its items along the $x$ and/or $y$ axis accordingly to update the layout.

5. **(Optional) Layout optimization**: If the layout optimization module is enabled, then after applying the navigation-based adjustments, invoke the layout optimizer on the updated layouts to further repair physical/constraint violations and write the optimized layouts back to the scene directories.

6. **Stopping criterion**: If $t = T$ and $U_t > 0$, a final navigation analysis can be executed to obtain the terminal reachability metrics, and the full iterative history is recorded.

**Reachability Metrics and Convergence Criterion**    Let $\mathcal{P}$ denote the set of all navigation paths to be evaluated over all scenes, and let

$$N = |\mathcal{P}| \tag{12}$$

be the total number of paths. For each path $p \in \mathcal{P}$, define its reachability label as

$$\text{reachable}(p) = \begin{cases} 1, & \text{if the path is reachable,} \\ 0, & \text{if the path is unreachable.} \end{cases}$$

The set of unreachable paths is then

$$\mathcal{P}_{\text{unreach}} = \{\, p \in \mathcal{P} \mid \text{reachable}(p) = 0 \,\}, \tag{13}$$

and the number of unreachable paths is

$$U = |\mathcal{P}_{\text{unreach}}|. \tag{14}$$

In implementation, we use the unreachable rate (in percentage form) as a global convergence metric:

$$r = \begin{cases} \dfrac{U}{N} \times 100\%, & N > 0, \\ 0, & N = 0. \end{cases} \tag{15}$$

The basic convergence criterion of the algorithm is

$$U = 0 \;\Rightarrow\; \text{stop}, \tag{16}$$

i.e., the iterations terminate once all paths are reachable. Optionally, more relaxed stopping conditions can be used in practice, for example, when the improvement between two consecutive iterations is small,

$$|r_t - r_{t-1}| < \varepsilon, \tag{17}$$

or when the number of unreachable paths no longer decreases over several iterations, or when the number of iterations exceeds three.

**Position Adjustments (Translation Updates)** For any object (e.g., a table, device, or platform) $o$, let its 2D position vector be

$$\mathbf{p} = (x, y)^\top. \tag{18}$$

For each unreachable path, the navigation analysis module outputs a translation suggestion of the form

$$(o, d, \Delta), \quad d \in \{x, y\},$$

meaning that object $o$ should be translated by $\Delta$ along direction $d$. If the suggestion is to translate along the $x$ axis, the update rule is

$$x' = x + \Delta, \qquad y' = y. \tag{19}$$

If the suggestion is to translate along the $y$ axis, the update rule is

$$x' = x, \qquad y' = y + \Delta. \tag{20}$$

In the implementation, for each platform object $o$, we first construct a mapping from the platform to the set of desktop items placed on it, denoted by $\mathcal{I}(o)$, based on the `initial_location` field in the layout. When the platform $o$ is translated, all objects $i \in \mathcal{I}(o)$ on that platform are translated by the same displacement, in order to preserve their relative geometric configuration:

$$\forall i \in \mathcal{I}(o): \quad \mathbf{p}'_i = \mathbf{p}_i + (\Delta x, \Delta y)^\top, \tag{21}$$

where $(\Delta x, \Delta y)$ is derived from the 1D translation rule above (either along $x$ or along $y$).

**Rotation Fixes (Rigid-Body Rotation Updates)** For platforms or devices that are classified as facing a wall, let their center position be

$$\mathbf{c} = (c_x, c_y)^\top, \tag{22}$$

their current orientation angle (around the $z$ axis, in degrees) be $\theta$, and the suggested target orientation from navigation analysis be $\theta^*$. The rotation increment is

$$\Delta\theta = \theta^* - \theta. \tag{23}$$

To use a rotation matrix, we convert this increment to radians:

$$\alpha = \Delta\theta \cdot \frac{\pi}{180}. \tag{24}$$

The corresponding 2D rotation matrix is

$$\mathbf{R}(\alpha) = \begin{bmatrix} \cos\alpha & -\sin\alpha \\ \sin\alpha & \cos\alpha \end{bmatrix}. \tag{25}$$

The platform orientation is updated to

$$\theta' = \theta^*. \tag{26}$$

For any object $i \in \mathcal{I}(o)$ placed on this platform, let its original world coordinates be

$$\mathbf{p}_i = (x_i, y_i)^\top, \tag{27}$$

and its offset from the platform center be

$$\mathbf{d}_i = \mathbf{p}_i - \mathbf{c}. \tag{28}$$

The new world coordinates after rigid-body rotation are given by

$$\mathbf{p}'_i = \mathbf{c} + \mathbf{R}(\alpha)\,\mathbf{d}_i. \tag{29}$$

If the object $i$ has its own orientation angle $\phi_i$ (in degrees), it is updated in sync with the platform, with a modulo $360°$ operation in implementation:

$$\phi'_i = (\phi_i + \Delta\theta) \bmod 360°. \tag{30}$$

This update ensures that the platform and all objects on it are rotated as a rigid body, thus correcting the device orientation while preserving the internal layout structure.

**Deduplication Strategy for Adjustments**   Navigation analysis may output multiple adjustment suggestions for the same object and direction, coming from different unreachable paths. To simplify updates and improve convergence, we perform deduplication: for each object $o$ and direction $d \in \{x, y\}$, we keep only the suggestion with the largest absolute translation magnitude.

Let the set of all raw suggestions be

$$\mathcal{A} = \{a_k\}_{k=1}^K, \quad a_k = (o_k, d_k, \Delta_k),$$

where $o_k$ denotes the object, $d_k \in \{x, y\}$ the direction, and $\Delta_k$ the translation distance. For a given $(o, d)$, define the subset

$$\mathcal{A}(o, d) = \{ a_k \in \mathcal{A} \mid o_k = o,\ d_k = d \}. \tag{31}$$

We then select

$$a^*(o, d) = \arg\max_{a \in \mathcal{A}(o,d)} |\Delta(a)|, \tag{32}$$

i.e., among all suggestions for the same object and direction, we retain the one with the largest absolute translation. The final set of executed adjustments is

$$\mathcal{A}^* = \{ a^*(o, d) \mid \exists\, a \in \mathcal{A}(o, d) \}. \tag{33}$$

This strategy avoids applying many small, potentially cancelling translations to the same object, and helps to achieve significant reachability improvement within a limited number of iterations.

In summary, Navigation Optimization follows the iterative framework above, which converts navigation unreliability feedback (including rotation fixes and position adjustments) into geometric updates of the scene layout, and optionally couples it with the layout optimizer for geometric and chemical repair. Through these iterations, the unreachable rate $r_t$ is progressively reduced until the predefined convergence criterion is satisfied.

### C.5. Semantic and Safety Priors for Functional Zoning

- **Safety-driven zoning.** Hazard attributes (e.g., volatile/toxic, flammable, heat-source) combined with protocol cues guide functional zoning and placement, e.g., volatile/toxic reagents are prioritized near/inside fume hood areas; flammables maintain separation from heat sources.

- **Workbench assignment pool.** From the asset library we form a *floor-level asset pool* and annotate which objects provide valid work surfaces that can host desktop items.

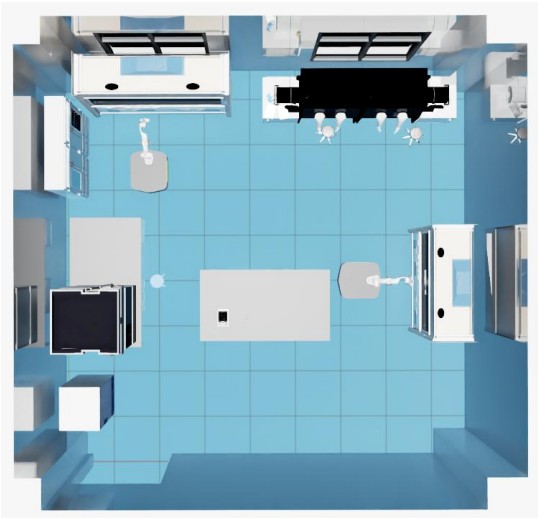 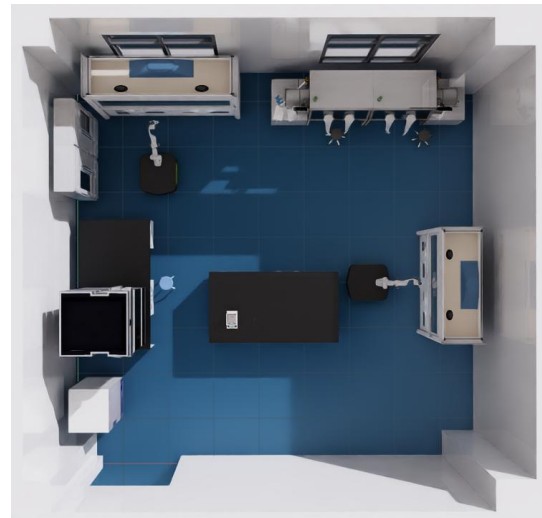

*Figure D.2.* Example illustration of navigation target pose generation (including offset inflation and heading) for a mobile manipulator in the laboratory scene.

## D. LabGen: Navigation Module Implementation Details

### D.1. Target Identification and Parent Object Resolution

First, the system extracts the target objects requiring navigation from the experimental protocol and locates their corresponding `.json` configuration files within the asset location information. These configurations contain placement details, including position coordinates (center point) and orientation. For each target object, the system extracts its bounding box (bbox) information.

When a target object's (e.g., a balance) `initial_location` is not set to `"floor"`, it indicates placement upon a parent object (e.g., a laboratory bench). In such cases, the system automatically identifies the parent object and compares the bounding box areas of the child and parent. If the parent object's bbox is larger, the navigation target is automatically replaced by the parent object. This strategy prevents the robot from attempting to navigate into the interior of a table to reach the center of a desktop asset, instead guiding it to a reachable position at the table's edge.

### D.2. Navigation Pose Generation

Upon determining the final target object, the system generates a specific navigation pose based on the object's geometry. Let the center point of the target object be $(x, y)$, its orientation be object_rz, and its dimensions be $(d_x, d_y)$. By adding the half-width $d_y/2$ to the center point, we obtain the object's boundary position. Defining the distance between the navigation point and the object boundary as dist, and introducing an inflation parameter offset_radius, the initial navigation distance is calculated as:

$$\text{dist} = \frac{d_y}{2} + 2 \times \text{offset\_radius}. \tag{34}$$

Here, one offset_radius accounts for the object boundary inflation, and the other for the robot's own safety inflation. If the target object has a rotation angle $r_z$ in the scene, the system calculates the offset relative to the object center:

$$dx_{\text{obj}} = -\text{dist} \cdot \sin(r_z), \quad dy_{\text{obj}} = -\text{dist} \cdot \cos(r_z). \tag{35}$$

Accordingly, the final navigation target coordinates are derived:

$$\text{Target\_X} = x + dx_{\text{obj}}, \quad \text{Target\_Y} = y + dy_{\text{obj}}. \tag{36}$$

The robot's orientation at this navigation point is set to ensure it faces the target object for operation:

$$\text{robot\_theta} = (\text{object\_rz} + 180) \bmod 360. \tag{37}$$

You can see some examples in Figure D.2.

## D.3. Sequence Generation and Optimization

The system generates a list of navigation targets following the sequence of steps in the experimental protocol. For two adjacent steps, the navigation point generated by the $i$-th step serves as the start point, and that of the $(i + 1)$-th step as the end point. If the start and end points for consecutive steps effectively coincide in space (e.g., "pick up beaker" immediately followed by "pour liquid," where the robot does not need to move), the system detects a zero-length path segment and discards it to avoid generating invalid navigation tasks. The final generated navigation points are represented in the form (Target_X, Target_Y, robot_theta).

## D.4. Dataset Configuration Format

The following JSON snippet illustrates the metadata structure for our tabletop asset placement:

---

### *Case Goal Pair Semantic & Navigation Specification*

**Specification ID:** `goal_pairs_case_01`
**Specification Type:** Navigation Goal Pair Configuration

**[Semantic Description]**
**Purpose:**
This specification defines a set of ordered navigation goal pairs, where each pair consists of a start pose and a corresponding end pose. These goal pairs are used to evaluate navigation, motion planning, or task execution performance in a structured environment.

**[Pose Representation]**
Each pose is represented as a 3-tuple:

$$(x, \ y, \ \theta)$$

where $x$ and $y$ denote planar position coordinates, and $\theta$ denotes the heading angle in radians.

**[Goal Pair Definitions]**
**Goal Pair 1:**

- **Start Pose:** $(1.211, \ 6.704, \ 1.571)$

- **End Pose:** $(2.100, \ 6.905, \ 0.000)$

**Goal Pair 2:**

- **Start Pose:** $(2.100, \ 6.905, \ 0.000)$

- **End Pose:** $(6.779, \ 4.000, \ 4.712)$

**Goal Pair 3:**

- **Start Pose:** $(6.779, \ 4.000, \ 4.712)$

- **End Pose:** $(2.423, \ 4.000, \ 1.571)$

**[Configuration Metadata]**

- Number of Targets: **10**

- Number of Goal Pairs: **3**

**[Intended Usage]**
This goal pair configuration is intended for benchmarking sequential navigation performance across multiple spatially distributed targets. The ordered structure enforces continuity between successive navigation tasks, enabling evaluation of path planning consistency and orientation handling.

---

**Notes:**
The numerical values are provided in continuous space and assume a consistent global coordinate frame. Orientation angles are expressed in radians and follow a standard planar navigation convention.

**D.5. 2D Navigation Modeling and Feasibility Checks**

After target generation, the system performs 2D navigation modeling. Given that robot movement in the laboratory is primarily planar, the problem is simplified to three parameters $(x, y, \theta)$. The system adds physical collision attributes to obstacles in the scene's `.usd` file and utilizes the `omni.tools.omap` method to convert the scene into a binary obstacle map (black representing obstacles, white representing traversable areas). Subsequently, continuous coordinates from the simulation environment are mapped to pixel coordinates to locate navigation points within the obstacle map.

For path planning, the robot's initial collision radius is set to $0.3\,\text{m}$. By inflating obstacle boundaries, the robot is treated as a particle, allowing the use of the A* algorithm on the inflated map. **Success Case:** If initial planning succeeds, the system outputs the corresponding navigation path. **Failure Case:** If planning fails, the system reads layout and object dimension data to reconstruct the environment on a 2D plane. It expands all object bounding boxes outward by the robot's physical radius, thereby simplifying complex volumetric collision problems into geometric "point-in-region" detection.

Based on this reconstruction, the system sequentially checks for specific failure modes:

1. **`start_blocked`:** Whether the robot's starting point lies inside an inflated obstacle (calculating penetration depth).

2. **`end_blocked`:** Whether the navigation target point is obstructed by tables or walls.

3. **`path_blocked`:** Whether any waypoints on the planned path traverse obstacles.

These results are consolidated into a structured output and, if necessary, forwarded to the Large Language Model for region-level judgment. Finally, these results are returned to the asset location information file to trigger the regeneration of navigation points and the execution of subsequent task planning.

# E. LabTouchstone: Evaluator Scoring and Report Schema

The evaluation framework $E_t$ provides a unified quantitative assessment of the generated laboratory layouts. The scoring logic integrates physical feasibility, chemical safety, and semantic and workflow consistency.

- **Report structure.** $E_t$ is emitted in a unified JSON containing an overall score, sub-scores, a violation list (with quantitative diagnostics like current pose and overlap degree), and improvement suggestions.

- **Score composition.** The total score $S_{total} = S_{phys} + S_{chem} + S_{consist} = 100$. Physical feasibility ($S_{phys}$) and Chemical safety ($S_{chem}$) are capped at 35 points each, while Semantic/Workflow consistency ($S_{consist}$) is capped at 30.

- **Physical feasibility sub-constraints.** We assess (i) tabletop boundary, (ii) desktop collisions, (iii) height consistency, and (iv) room-level constraints. Any geometric violation triggers the **Geometry Violation Penalty**, setting the satisfaction of involved chemical constraints to 0.

- **Chemical safety as continuous satisfaction.** We instantiate constraints relevant to the protocol and compute a satisfaction score $s \in [0, 1]$ per instance.

- **Semantic and workflow consistency.** We employ a Vision-Language Model (VLM) to evaluate (i) realism, (ii) functionality, (iii) layout reasonableness, and (iv) completion. The evaluation can be toggled between *strict*, *medium*, and *lenient* modes to adapt to different assessment rigors.

**E.1. Chemical Safety**

$S_{chem}$ (35 pts) evaluates compliance with dynamic safety constraints triggered by protocol-specific hazards (e.g., toxicity). It aggregates four sub-metrics using "Worst 3 Average" for critical risks and standard averaging for general requirements. To ensure realizability, a Geometry Violation Penalty ($\nVdash_{geo}$) is applied, assigning zero safety satisfaction to any physically unstable asset.Figure E.3: Visualization of LabTouchstone evaluation metrics via a donut chart.

The detailed formulation for each metric is provided below:

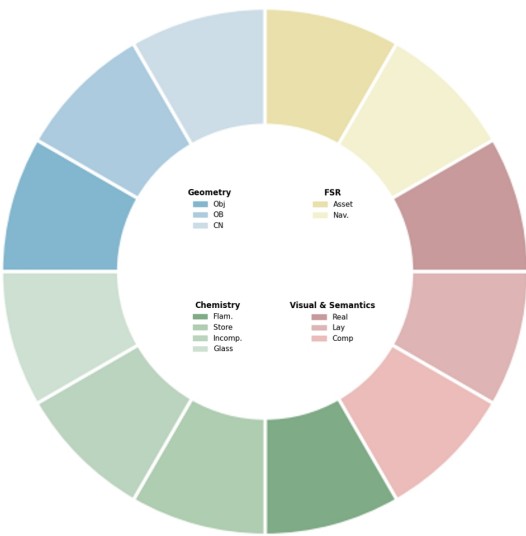

*Figure E.3.* Overview of our LabTouchstone. We present four primary categories: Geometry, FSR, Chemistry, and Visual & Semantics, with their respective sub-metrics detailed in the legend.

### E.1.1. METRIC 1: FLAMMABLE REAGENT AND HEAT SOURCE SEPARATION

Evaluates the safety distance between flammable reagents and heat sources. For each pair $i$, the satisfaction $s_i$ is a piecewise mapping based on actual distance $d_i$ and threshold $d_{min}$:

$$s_i = \mathbb{1}_{geo}(i) \cdot \begin{cases} 1, & \text{if } d_i \geq d_{min} \\ \frac{d_i}{d_{min}}, & \text{if } d_{low} \leq d_i < d_{min} \\ 0, & \text{if } d_i < d_{low} \end{cases} \tag{38}$$

We apply the **Worst 3 Average Strategy** to prioritize critical risks:

$$S_{\text{flammable}} = \frac{1}{\min(n,3)} \sum_{j=1}^{\min(n,3)} s_{(j)} \tag{39}$$

### E.1.2. METRIC 2: REAGENT STORAGE IN CABINETS

Ensures reagents are stored in designated cabinets rather than exposed on workbenches. Satisfaction for reagent $i$ is binary based on containment:

$$s_i = \mathbb{1}_{geo}(i) \cdot \mathbb{1}_{inside}(i) \tag{40}$$

The category score uses the **Average Strategy**: $S_{\text{storage}} = \frac{1}{n} \sum_{i=1}^{n} s_i$.

### E.1.3. METRIC 3: INCOMPATIBLE REAGENT SEPARATION

Identifies incompatible reagent pairs via a chemical property database and enforces safety buffers. Similar to Metric 1, it utilizes a distance-based piecewise mapping and the **Worst 3 Average Strategy**:

$$S_{\text{incompatible}} = \frac{1}{\min(n,3)} \sum_{j=1}^{\min(n,3)} s_{(j)} \tag{41}$$

where $s_{(j)}$ are the lowest satisfaction scores in the category.

E.1.4. METRIC 4: GLASS EQUIPMENT EDGE AVOIDANCE

Minimizes breakage risk by keeping fragile equipment away from workbench edges. The individual safety score $s_i$ for an asset $i$ is calculated based on the ratio of its distance to the nearest edge relative to a safety threshold:

$$s_i = \min\left(1, \frac{d_i}{D_{safe}}\right) \tag{42}$$

Where:

- $d_i$ denotes the shortest Euclidean distance from the asset $i$ to the nearest workbench edge.

- $D_{safe}$ represents the predefined safety distance threshold.

The final category score follows the **Average Strategy**:

$$S_{\text{glass}} = \frac{1}{n} \sum_{i=1}^{n} s_i \tag{43}$$

**E.2. Semantic and Workflow Consistency**

To quantitatively evaluate the semantic quality of the generated laboratory scenes, this work designs a semantic scoring system based on Vision-Language Models (VLMs). For each scene, we first render or collect a set of laboratory images. These images, along with optional experimental protocol information, are fed into the VLM, which then outputs structured scoring results based on unified evaluation criteria.

E.2.1. SEMANTIC DIMENSIONS AND SCORE DEFINITIONS

The semantic evaluation comprises three dimensions, each scored on a scale of 0–10:

- **Realism**: Measures whether the scene conforms to the scale, equipment types, and overall appearance of a real chemical laboratory—i.e., "does it look like an actual laboratory?" High scores (8–10) require the layout to follow real laboratory norms with reasonable equipment types and no "amateurish" placement. Medium scores (4–7) indicate basically correct equipment but slight irrationalities in scale or positioning. Low scores (0–3) correspond to the presence of unreasonable or hazardous elements or layouts that clearly violate experimental common sense. Specifically, if the equipment combination significantly violates laboratory conventions, a hard constraint of 0 is enforced.

- **Layout Reasonableness**: Evaluates whether the spatial arrangement meets safety and operational convenience requirements. High scores require clear traffic paths, logical placement of hazardous equipment, and sufficient operational space. Medium scores correspond to local congestion or suboptimal workflow flow. Low scores correspond to severe occlusion, collision risks, or the mixing of hazardous and general-purpose areas. If layout issues seriously impede safety or experimental feasibility, the constraint of 0 is applied.

- **Completion**: Measures whether the scene is visually "complete," including whether the configuration of primary and auxiliary equipment is sufficient. High scores require primary and auxiliary equipment to be largely complete, with workbenches and walls not appearing overly vacant. Medium scores indicate primary equipment is present but auxiliary equipment is lacking. Low scores correspond to sparse scenes or a distinct "unfinished" feel. If over 80% of the workbench or wall space is nearly empty, the constraint of 0 is applied.

For each scene, these four dimensions are treated as equally weighted indicators. The total score is defined as:

$$S_{\text{total}} = s_{\text{realism}} + s_{\text{layout}} + s_{\text{comp}}, \tag{44}$$

ranging from 0 to 30. We also define the average score:

$$\bar{S} = \frac{S_{\text{total}}}{3}, \tag{45}$$

which serves as a brief representation of the overall semantic quality.

### E.2.2. SCORING PROMPTS AND PROTOCOL-CONDITIONED CONSTRAINTS

To obtain stable and consistent semantic ratings, we constructed detailed scoring prompts for the VLM. The prompt first provides textual definitions for the four dimensions and gives typical criteria for "High (8–10)," "Medium (4–7)," and "Low (0–3)" scores, alongside a "Key Judgment Rule" for each dimension, requiring the model to explicitly reference these thresholds and rules during scoring.

When the experimental protocol associated with a scene is available, we incorporate the experiment name, a brief introduction, and the required asset list (including names and purposes of various instruments and reagents) into the prompt. The model is instructed to first understand the specific experimental task served by the scene and then judge whether the images possess the critical equipment and spatial configurations necessary to complete said experiment, adjusting the "Functionality" and "Completion" scores accordingly. The prompt explicitly states that "all assets listed in the protocol are available in the images" to ensure the model checks the scene configuration against the protocol requirements.

### E.2.3. SCORING MODES AND RIGOR CONTROL

To adapt to different requirements for subjective scoring rigor across various experimental settings, we introduced three scoring modes in the prompt: *strict*, *medium*, and *lenient*. These modes share the same semantic definitions and score ranges but constrain the "style" of the scoring through additional natural language rules:

- **Strict Mode**: Requires the model to *strictly* enforce the hard constraints for key rules (e.g., missing critical equipment or severe safety hazards) and significantly penalize any equipment deficiency or safety risk. The overall scoring is conservative.

- **Medium Mode**: Requires the model to allow slight deviations while observing the key rules; if the scene "basically meets experimental needs" but only lacks a few minor items, a medium-to-high score can still be given. This mode strikes a balance between strictness and tolerance and is the default setting used in this paper.

- **Lenient Mode**: Directs the model to focus on "overall feasibility." As long as the scene possesses most critical equipment and can reasonably complete the core experimental steps, a score higher than 8 may be awarded. It does not excessively penalize the absence of auxiliary equipment and is suitable for cases where visual diversity is of greater concern.

These modes are implemented by appending different natural language instructions at the end of the prompt, thereby applying soft constraints on the LLM's scoring strategy without changing the interface or data structure.

### E.2.4. INFERENCE AND RESULT PARSING

At the implementation level, for each scene, we first collect all image files in the scene directory, encode them as base64 image data, and send them as multimodal input along with the textual prompts to the VLM. The model is required to return scores and textual explanations for the three dimensions in a strict JSON structure, e.g., `{"realism": {"score": 8, "reason": "..."} , ...}`. The program-side parser extracts the scores to calculate $S_{\text{total}}$ and $\bar{S}$. During batch evaluation, means and distributions of these metrics across the scene collection can be statistically analyzed for subsequent comparative experiments and ablation studies.

## F. Qualitative Failure Case Analysis

To provide a deeper understanding of baseline limitations, we present representative failure cases from Holodeck and SceneWeaver with explicit violation annotations. These examples correspond to the quantitative metrics reported in Table 2 and Table 3.

These visualizations emphasize four typical failure modes observed across baselines:

1. **Asset insufficiency:** required instruments or reagents are missing or mismatched.

2. **Geometric violations:** collisions and out-of-boundary placements.

3. **Chemical safety violations:** flammable or incompatible chemicals improperly positioned.

**SceneWeaver**  **Holodeck**

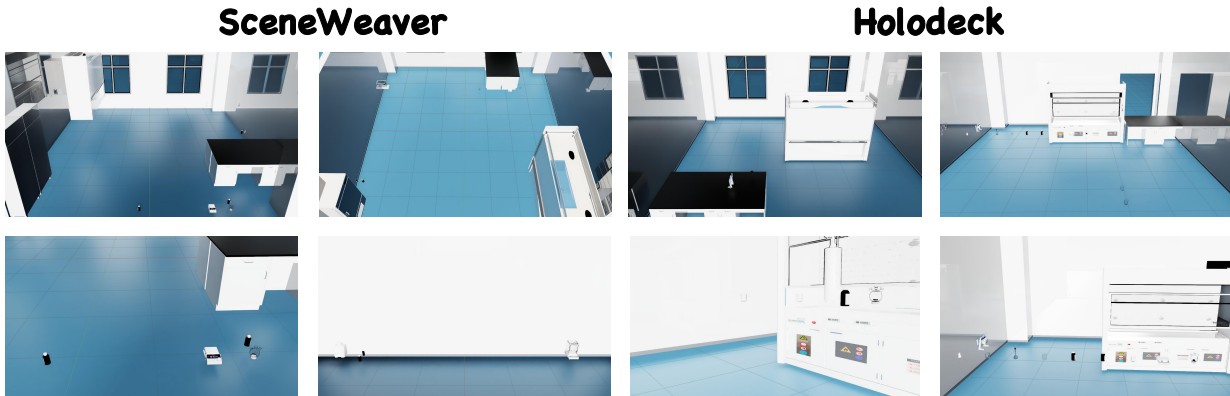

*Figure E.4.* Representative failure cases for Holodeck and SceneWeaver. Red bounding boxes indicate collisions, out-of-boundary placements, and unsafe chemical arrangements.

4. **Workflow and navigation failures:** missing assets or improper placement impede multi-step protocol execution.

