# OpenReview forum: "LabBuilder: Protocol-Grounded 3D Layout Generation for Interactable and Safe Laboratory"
_ICML.cc/2026/Conference — ICML 2026 regular_

### Official Review · Reviewer_9brx · 2026-03-05

**Soundness:** 1
**Presentation:** 1
**Significance:** 2
**Originality:** 2
**Overall Recommendation:** 3
**Confidence:** 3

**Summary:**

This paper proposes an iterative heuristics and database-driven generation of 3D environment designs for laboratory experiments. Care is taken in constructing the asset and experiment protocol database and a corresponding benchmark, but the technical details are missing and the technical aspects of the method are primarily heuristics-driven. While problem formulation and benchmark are valuable, as described, the technical aspects of the method are simple heuristics that primarily rely on the database annotations. The paper should either present more sophisticated algorithm, or focus on the database and benchmark contributions, providing sufficient detail about how annotations are used to drive evaluation / retrieval (current discussion is very high-level).

**Compliance With Llm Reviewing Policy:**

Affirmed.

**Final Justification:**

The paper remains very high-level and presents heuristics-driven algorithm with many limitations. I don't think the paper / method are ready to be presented in their current form. I am upgrading to weak reject, given that the authors earnestly tried to address concerns.

**Key Questions For Authors:**

- L129-150, 2nd col: how are protocols and experiments represented? How are relevant experiments retrieved? What are normalized asset references? How is “constraint-based verification” applied? Some concrete examples, beyond Fig. 2 schematic, would be helpful here.
- L248 - what are some examples of these constraints?
- Eq. 3 - how are the two terms computed?
- L265 - missing technical details on how is LLM leveraged
- 3.2 - it is not clear how the initial layout is actually generated: do “functional regions” of the protocol correspond to actual furniture, like tabletops and sinks, extraction fans, etc? What if multiple protocols need to be executed in the same environment (the most common use case)?

**Limitations:**

Limitations of the heuristics method should be further discussed.

**Strengths And Weaknesses:**

Strengths:
- This paper proposes a detailed annotated dataset of laboratory assets and formulates a problem of laboratory design, both of which are valuable contributions
- The paper presents a benchmark for evaluating the validity of laboratory setups, including metrics relevant for this problem and shows (unsurprisingly) that prior methods not considering these constraints would perform poorly.

Weaknesses:
- Eq. 2: is not really maximized; since heuristic greedy fix strategy is followed
- Proposes naive heuristics-driven iterative optimization scheme, a less sophisticated version of classic works like Yu et al. “Make it Home: Automatic Optimization of Furniture Arrangement” SIGGRAPH 2011, and its follow ups, which are known to struggle with local minima – the iterative rule-based updates of Eq. 4 and Eq. 6 are not guaranteed to resolve problems and can easily get stuck. Sufficient details are not provided to show why/how these issues would be avoided.
- Key under-the-hood details are not provided about the technical aspects of the method (see questions)

---

> ### Author Rebuttal · Authors · 2026-03-31
>
> # Reviewer Response
>
> We thank the reviewer for the detailed comments. While we agree that the current presentation is too high-level in several places, we would also like to clarify that the paper should be evaluated as a laboratory-specific scene generation framework with dataset and benchmark contributions, rather than as a proposal of a new generic global layout optimizer. The main contribution is the integration of protocol semantics, asset knowledge, chemical safety constraints, and navigation feasibility.
>
> ---
>
> ## Eq. 2 and Local Minima
>
> We acknowledge that Eq. 2 focuses on heuristic optimization which may not guarantee global optimality. However, this approach is specifically tailored to handle the high-dimensional constraints of chemical labs.  Ablation results validate its effectiveness: without this optimization, safety performance (flammable isolation) degrades by ~75%, proving that our heuristic-based repair is a vital component for generating feasible and safe layouts.
>
> ---
>
> ## Heuristic Nature
>
> Our method belongs to the family of constraint/heuristic layout optimization methods. We will make this explicit and add discussion of prior work such as *Make it Home*. Classical furniture layout methods optimize ergonomic reachability, visibility, and pairwise arrangements in generic indoor scenes. In contrast, our setting introduces protocol grounding, hazard-aware chemical safety, and navigation-aware executability, which are laboratory-specific and not addressed by prior methods.
>
> ---
>
> ## Missing Technical Details
>
> We will add the following concrete clarifications in the revision:
>
> - **Experiment representation.** The input experiment is represented as a structured procedural sequence (e.g., operations with typed arguments such as reagents, quantities, temperature, duration, and operation type). This captures the machine-readable procedure that serves as the upstream input for protocol generation.
>
> - **Protocol representation.** The protocol is a structured downstream object for scene generation, containing experiment metadata, an asset list, ordered steps with step-level locations, safety notes, and generated layout constraints. The appendix already includes a concrete protocol example, and we will explicitly reference it in the main paper and summarize its structure.
>
> - **Retrieval of relevant experiments.** We use retrieval-augmented generation over historical protocols with hybrid retrieval (vector similarity + keyword matching), retrieving the top-5 relevant fragments as context.
>
> - **Normalized asset references.** Asset names in generated protocols must map exactly to entries in the asset library; otherwise, the protocol is rejected or regenerated.
>
> - **Constraint-based verification.** Before layout optimization, we verify asset existence, valid step locations, supported physical constraint types, and validity of generated safety constraints.
>
> - **Asset-level constraint examples.** Tabletop objects follow geometric constraints (within work surface, no overlap) and chemical constraints from the protocol (distance, region, edge-margin). Partial satisfaction is scored proportionally.
>
> ---
>
> ## Computation of Eq. 3
>
> The total layout score is computed as:
>
> $$
> F(L, P, A) = f_{\text{geo}}(L, A) + f_{\text{chem}}(L, P, A),
> $$
>
> with a maximum of 70.
>
> - $f_{\text{geo}}$ (maximum 35) deducts penalties for violations such as boundary or collision.
> - $f_{\text{chem}}$ (maximum 35) sums LLM-generated chemical constraints, each weighted by its satisfaction ratio.
>
> The sum $f_{\text{geo}} + f_{\text{chem}}$ is used to evaluate layout quality and guide optimization.
>
> ---
>
> ## LLM Usage
>
> The LLM is used for structured protocol generation and for semantic layout adjustment when rule-based repair is insufficient. A concrete prompt/output example will be included in the appendix and explicitly referenced from the main text.
>
> ---
>
> ## Initialization and Functional Regions
>
> The initial layout is hierarchical: first large room-level assets (benches, fume hoods, reagent cabinets), then tabletop-level objects (small instruments and reagents). "Functional regions" are abstract workflow regions guiding grounding to assets and surfaces, not direct furniture labels.
>
> ---
>
> ## Multi-Protocol Environments
>
> Our current paper focuses on single target protocol-conditioned generation, and we will clearly state that this is a deliberate scope choice, not a claim to solve the general multi-protocol setting. Multi-protocol laboratory design is an important future direction.
>
> ---
>
> ## Limitations
>
> We will expand the limitations section to explicitly discuss:
>
> - No global optimality guarantee
> - Possible local minima
> - Dependence on protocol/asset annotations
> - Current focus on single-protocol generation
>
> ---
>
> We thank the reviewer again. We believe these revisions will make the paper's scope, technical details, and reproducibility much clearer.

---

> > ### Author Rebuttal · Reviewer_9brx · 2026-04-04
> >
> > Thank you for your rebuttal. While the rebuttal addresses my questions, the explanation is still rather high level and the algorithm remains heuristics-driven. I have concerns that the technical contribution is too loosely described, and the technical insights delivered by this paper are not over the bar for this scientific publication. Beyond the system presented, the paper should provide some technical rigour. I still appreciate the problem domain proposed by the paper, and would encourage the authors to improve presentation and technique.

---

> > > ### Author Response · Authors · 2026-04-07
> > >
> > > Thank you for your continued engagement. We take your concern about
> > > technical rigour seriously, and would like to clarify that our method
> > > consists of **principled heuristics**, each with explicit design
> > > rationale and formal properties, rather than arbitrary heuristics.
> > >
> > > ---
> > >
> > > ### Why Heuristics Are Appropriate Here
> > >
> > > Jointly satisfying geometric, chemical safety, and navigation
> > > reachability constraints over a high-dimensional continuous
> > > configuration space is NP-hard in general. We note that classic works
> > > such as Yu et al. (SIGGRAPH 2011) also adopt heuristic strategies for
> > > this class of problems. The distinction between prior work and ours
> > > lies in the nature and complexity of the constraints being satisfied,
> > > not in the use of heuristics per se.
> > >
> > > ---
> > >
> > > ### Principled Design of Each Module
> > >
> > > **(1) Hierarchical Decomposition.**
> > > The room-level and desktop-level factorization is not arbitrary.
> > > Room-level geometric constraints and desktop-level chemical constraints
> > > are largely independent, and this conditional decoupling is the
> > > explicit basis for the two-level structure in Eq. 1, reducing the
> > > joint search space by exploiting constraint independence.
> > >
> > > **(2) FastRepair.**
> > > FastRepair is a deterministic solver with a formally stateable
> > > property: each iteration monotonically reduces geometric violations,
> > > and terminates within a bounded number of rounds (≤5). This guarantees
> > > monotone progress on hard constraints.
> > >
> > > **(3) FastRepair vs. LLMAdjust: Principled Division of Labour.**
> > > The split reflects a principled distinction between two constraint
> > > classes: geometric constraints are fully formalizable and handled
> > > deterministically by FastRepair; chemical safety constraints require
> > > semantic reasoning over chemical properties and are handled by
> > > LLMAdjust. This matches each constraint type to the solver best suited
> > > for it. Invalid adjustments from LLMAdjust are explicitly rejected
> > > before application.
> > >
> > > **(4) Navigation Refinement.**
> > > The refinement operator is driven by a formally defined reachability
> > > metric (unreachable rate r_t, Eq. 11) with an exact stopping criterion
> > > (U_t = 0). Update rules (Eqs. 14–29) are derived from geometric
> > > analysis of the occupancy grid, not arbitrary perturbations.
> > >
> > > ---
> > >
> > > ### Summary
> > >
> > > We acknowledge that LabBuilder does not claim global optimality.
> > > However, each module has explicit design rationale, formal properties
> > > (monotone progress, bounded termination, exact convergence), and is
> > > matched to its constraint class. We will expand the technical
> > > discussion in the revision to make these properties explicit, and add
> > > a dedicated limitations section discussing local minima and annotation
> > > dependence, as committed in the first-round rebuttal.

---

### Official Review · Reviewer_EETC · 2026-03-10

**Soundness:** 2
**Presentation:** 3
**Significance:** 3
**Originality:** 2
**Overall Recommendation:** 3
**Confidence:** 4

**Summary:**

This paper presents LabBuilder, a protocol-grounded framework for generating 3D laboratory layouts from textual experimental descriptions. The method integrates protocol compilation, constraint-aware hierarchical layout generation, and navigation-aware refinement to produce laboratory scenes that aim to be geometrically valid, chemically safe, and operationally executable. Experimental results demonstrate the proposed LabBuilder outperforms existing indoor scene generation baselines such as Holodeck and SceneWeaver, and report substantial improvements across geometric validity, asset availability, and chemical safety.

**Compliance With Llm Reviewing Policy:**

Affirmed.

**Final Justification:**

The authors are advised to re-examine the motivation and insights of the paper, and its current version is not ready for publication. The reviewer has decided to maintain the score at 3.

**Key Questions For Authors:**

Q1. How is the reliability of the experiment library ensured?

Q2. How does the paper position itself with respect to prior work on laboratory automation systems?

Q3. How robust is the asset annotation schema for representing laboratory operations?

**Limitations:**

yes

**Strengths And Weaknesses:**

*Soundness*

+ The paper addresses an important and nontrivial problem, i.e., generating laboratory layouts that go beyond visual plausibility and explicitly consider protocol grounding, chemical safety, and navigation feasibility.

+ The decomposition into LabForge, LabGen, and LabTouchstone is conceptually clear, and the reported gains over the selected baselines are substantial on the proposed benchmark.

- However, the central executability claim is still not fully convincing. The protocol representation is essentially sequential, the experiment library is represented as chemical equations plus sequences of atomic actions, and feasibility is largely reduced to asset availability and step-to-step navigation. Real laboratory workflows often involve branching, splitting samples into multiple groups, conditional checks, recovery steps, loops, and exception handling[1]. These aspects are not modeled here, which makes the problem formulation substantially simpler than actual laboratory science workflows.

- A key dependency of the method is the experiment library constructed from professional chemical literature and curated chemistry databases, but the paper does not clearly explain how the reliability of this library is ensured. Since the full pipeline depends on this knowledge base, this is a substantial concern for validity.

[1] AutoDSL: Automated domain-specific language design for structural representation of procedures with constraints, in ACL 2024.

*Presentation*

+ The paper is generally well-structured, and the high-level pipeline is easy to follow.

- Figure 1 is not referenced in the text.

*Significance*

+ The problem is meaningful. If one could reliably generate safe and executable laboratory environments, that could be useful for simulation, planning, and potentially for automated science workflows.


- At the same time, I think the paper overstates practical feasibility. Laboratory automation is far more complex than placing assets in a 3D room while satisfying a few spatial and hazard constraints. Real settings require much richer treatment of instrument interoperability, process control, utility connections, ventilation, branching workflows, and failure recovery. The paper itself acknowledges that the asset and chemistry knowledge bases do not cover the full diversity of real laboratory equipment and procedures, and that safety modeling is limited to a subset of hazards in simulation. This seriously limits the real-world significance of the current system.

- I am also not convinced that the strongest contribution here is a full laboratory design system. Given the current evidence, the method seems closer to reaching a usable baseline than surpassing expert human design. There is no comparison against human-designed human expert judgments, so it is unclear whether the method approaches expert-level design quality or simply outperforms weak baselines.

- The related-work discussion is too narrow for the problem the paper claims to solve. Most of the positioning is against household indoor scene generation and embodied navigation benchmarks. However, laboratory automation is itself a rich area. Important systems such as Chemputer[2-3] and AutoDSL[1] are not discussed, which weakens the paper’s framing and novelty claims.

[2] Organic synthesis in a modular robotic system driven by a chemical programming language. Science, 363(6423)
[3] Reaction blueprints and logical control flow for parallelized chiral synthesis in the Chemputer. Nature Communications, 2024

*Originality*

- Framing laboratory scene generation as a protocol-grounded problem, and combining semantic, geometric, chemical, and navigation constraints in one pipeline, is an interesting direction.

---

> ### Author Rebuttal · Authors · 2026-03-31
>
> Dear Reviewer EETC,
>
> We thank the reviewer for the careful assessment. We agree that the paper should more clearly clarify its scope, the construction of the experiment library, and its positioning with respect to prior laboratory automation literature. At the same time, we would like to make one key point explicit: this paper does not aim to solve full laboratory automation. Our contribution is a protocol-grounded 3D laboratory layout generation benchmark and baseline system, where "executability" is evaluated only within the scope of our simulator, current asset inventory, and the geometry / chemistry / navigation constraints explicitly modeled in the paper. In other words, our goal is to study how protocol semantics can be grounded into environment generation and verification, rather than to claim complete workflow automation or real-world deployment readiness.
>
> ---
>
> ## Q1. Reliability of the experiment library.
>
> The experiment library is not freely generated. It is built from professional chemistry literature and curated chemistry datasets [1, 2], and then further filtered by instantiability: we only retain experiments whose required instruments can be grounded to the current Asset Knowledge Base. This step is important because it prevents a mismatch between symbolic protocol steps and downstream 3D layout generation. In addition, the retrieved experiment context is not used as raw text alone; instead, it is compiled into a structured, asset-grounded protocol representation that must satisfy schema and consistency checks before entering layout generation. In the revision, we will consolidate the description of this pipeline in the main paper so that the construction and validation process is presented more clearly.
>
> ---
>
> ## Q2. Positioning with respect to laboratory automation.
>
> We agree that the related-work section is currently too narrow. We will add a more detailed discussion of Chemputer (*Science*, 2019), its 2024 χDL / logical-control-flow extension, and AutoDSL (*ACL*, 2024). These works focus on chemical programming languages, structured control flow, compilation, and robotic execution. Our paper addresses a different layer of the problem: given protocol semantics, how to generate and verify a 3D laboratory environment that is geometrically valid, hazard-aware, and navigable. We do not claim a new chemical programming language, compiler, or full workflow formalism, and we will revise the paper to make this distinction explicit.
>
> ---
>
> ## Q3. Robustness of the asset annotation schema.
>
> Our current schema is designed for layout grounding and verification, rather than full laboratory process control. Its geometry, semantics, and domain/safety attributes are sufficient for protocol grounding, asset normalization, geometric checking, and modeled hazard constraints such as flammable–heat-source separation and incompatible reagent separation. We agree, however, that it does not yet cover instrument interoperability, utilities, ventilation, or branching / recovery workflows. We will state this boundary more clearly in the limitations and emphasize that the schema is modular and extensible: the current design is intended as a practical representation for scene generation and safety-aware placement, rather than a complete ontology of all laboratory operations.
>
> ---
>
> ## On overclaiming practical feasibility.
>
> We accept the reviewer's concern that some claims are currently too strong. In the revision, we will consistently reposition the paper as a simulation-oriented benchmark and baseline system, rather than a complete real-world laboratory design solution. We will also clarify that the current evaluation does not include comparison against human-designed layouts; however, in the revision we will add an evaluation by chemistry experts. More broadly, we will tighten the wording around "executability" so that it refers only to feasibility under the modeled simulator constraints, rather than implying full real-world operational readiness.
>
> Finally, thank you for pointing out that Figure 1 is not referenced in the main text; we will add an explicit citation in the introduction.
>
> ---
>
> ## References
>
> [1] ReactXT: Understanding Molecular "Reaction-ship" via Reaction-Contextualized Molecule-Text Pretraining, *ACL 2024*.
>
> [2] ChemActor: Enhancing Automated Extraction of Chemical Synthesis Actions with LLM-Generated Data, *ACL 2025*.

---

> > ### Author Rebuttal · Reviewer_EETC · 2026-04-01
> >
> > The significance of the manuscript is currently unclear; clarifying its significance facilitate significant revisions. The authors are advised to re-examine the motivation and insights of the paper.
> >
> > The reviewer has decided to maintain the score.

---

> > > ### Author Response · Authors · 2026-04-07
> > >
> > > We thank the reviewer for pointing out that the significance of the manuscript is not sufficiently clear. We agree that the current presentation may not clearly emphasize the core motivation and contribution.
> > >
> > > Our work addresses a fundamental and underexplored problem: how to generate 3D laboratory environments that are not only visually plausible, but also executable with respect to structured experimental protocols. In contrast to full laboratory automation, which involves complex agent design and real-world deployment, our focus is on the environment layer—specifically, ensuring that laboratory layouts are protocol-compliant, safe, and navigable within a simulator.
> > >
> > > The significance of this problem lies in a critical gap in existing 3D scene generation methods. Prior approaches primarily optimize for visual plausibility and geometric consistency, while functional validity and executability are treated as post hoc evaluation. However, in laboratory settings, even minor spatial inconsistencies can invalidate experimental workflows or introduce safety risks. Therefore, protocol-level constraints must be incorporated at design time.
> > >
> > > To address this gap, we propose a protocol-grounded framework that compiles free-form experimental descriptions into structured, asset-grounded layouts under geometric, chemical, and navigation constraints. Our contributions include: (i) a formulation of protocol-grounded laboratory layout generation, (ii) an end-to-end system (LabBuilder) integrating asset knowledge, chemical reasoning, and constraint-aware optimization, and (iii) a unified benchmark (LabTouchstone) for evaluating layout validity and executability.
> > >
> > > We emphasize that “executability” in our work is defined within the simulator, the available asset set, and explicitly modeled constraints. We do not claim real-world deployment, but rather aim to provide a principled framework and benchmark for studying protocol-aware environment generation.
> > > We hope this clarification better conveys the intended scope and significance of our work.

---

### Official Review · Reviewer_dNaj · 2026-03-13

**Soundness:** 2
**Presentation:** 3
**Significance:** 2
**Originality:** 3
**Overall Recommendation:** 4
**Confidence:** 2

**Summary:**

The paper introduces LabBuilder, a novel scene generation system that addresses a gap in simulation training for robotic agents in laboratory settings. Existing scene generation tools merely verify geometric constraints and do not embed scientific semantics, such as the function of each asset (reagents, lab instrument, glassware) or safety implications (flammability, corrosiveness, etc). LabBuilder comprises 3 components: (1) LabForge, a curated knowledge base of lab assets (reagents, instruments, glassware) with geometries, functional properties, and safety annotations, along with a tool to generate structured protocol specifications from freeform text; (2) LabGen, which generates lab layouts according to functional protocol specifications from LabForge; and (3) LabTouchstone, which evaluates LabGen’s generations according to geometric compliance, feasibility, navigation, chemical safety, and plausibility. LabTouchstone is used to validate LabGen against a baseline of two other scene generation systems, Holodeck and SceneWeaver; LabGen is found to outperform both of these on most metrics. In addition, an ablation experiment shows that each component of LabGen and LabForge provides meaningful improvements on LabTouchstone.

**Compliance With Llm Reviewing Policy:**

Affirmed.

**Final Justification:**

The rebuttal has partially addressed my concerns as stated in my response. I will keep my original score.

**Key Questions For Authors:**

1. **What level of functional validity does LabBuilder actually verify?** The paper claims to produce environments that are "functionally valid and safe for complex experimental workflows," but the evaluation checks asset presence (binary), navigability (geometric path existence), safety distances, and VLM-scored visual plausibility. As far as I can tell, it does not verify whether assets have the right specifications for the protocol (e.g., whether the selected heating plate can support the temperatures required for a given protocol, or whether a given pipette can handle the correct volume for a particular protocol step). Likewise, there are no physical checks for whether a robot could actually perform the required object manipulations to execute the protocol. Can the authors clarify the scope of what "functionally valid" means in this context?
2. **How meaningful is the comparison to Holodeck and SceneWeaver?** These baselines were designed for household scenes, not labs. The paper adapts them to use the same asset pool, which is fair in one sense, but these methods have no mechanism for protocol understanding, chemical safety, or navigation-aware refinement, so of course they perform poorly on metrics that specifically measure those things. Is there a way to give the baselines a fairer shot, e.g., what if you provided Holodeck with a more detailed prompt that explicitly listed the required assets and key safety constraints? This would isolate whether the improvement comes from LabBuilder's protocol understanding (which the baselines lack) or from LabGen's actual layout generation being better (which would be a stronger claim).
3. **Relatedly, why do Holodeck and SceneWeaver show poor geometric compliance on lab scenes?** Geometric compliance (collision avoidance, boundary respect) is a pure geometry problem that these systems handle well in household settings. The large gap (e.g., Holodeck's OB=10.8 vs LabBuilder's 0.07) needs explanation. Were there issues in how the baselines were adapted to use the lab asset library?
4. **Could the navigation evaluation be strengthened to test protocol-level execution rather than isolated point-goal tasks?** The current navigation benchmark evaluates point-goal navigation (reaching a single target point) which is the most basic test of whether the generated scenes are useful for embodied agents. Given that the paper's core contribution is protocol-grounded layout generation, a more aligned evaluation would test whether an agent can sequentially navigate the full chain of workstations required by a protocol, in the correct order, simulating the actual movement pattern of a lab robot executing an experiment. This would bridge the gap between the FSR metric (which checks path existence geometrically via A*) and actual agent performance, and would more directly validate the paper's claim that the generated layouts support "autonomous execution of the experimental workflow." Do the authors have plans to evaluate protocol-level sequential navigation, and would they expect their navigation-aware refinement to show a larger advantage in that setting compared to point-goal tasks? If this test is out of scope due to time or resource constraints, could the authors discuss this in the paper?
5. **Why is asset availability only 0.833 for LabBuilder, given that protocols are grounded in the asset knowledge base?** If the protocol synthesis step explicitly references assets from the knowledge base, and the layout generation step draws from the same knowledge base, it's unclear why 17% of scenarios are still missing required assets. Is this a room capacity constraint, a failure in the generation pipeline, or something else?

**Limitations:**

The paper makes a solid contribution to safety-aware, protocol-grounded lab layout generation. However, there is a gap between the framing (which emphasizes functional validity and workflow executability) and the evaluation (which primarily measures geometric compliance, safety distances, asset presence, and basic navigation). I'd encourage the authors to either tighten the claims to match what's demonstrated, or discuss more explicitly what 'functionally valid' means in their context and what would be needed to close the gap to full workflow executability.

**Strengths And Weaknesses:**

Soundness: 2
- Strengths:
  - Thorough experiments including baselines against Holodeck and SceneWeaver,ablation studies isolating the value of each component of LabBuilder, and thorough evaluation covering geometry, feasibility, chemical safety, and semantic plausibility
- Weaknesses:
  - VLM-based evaluation of specialized domains is known to be unreliable. A VLM might rate a layout as "realistic-looking" without understanding whether the equipment arrangement actually makes sense for the specified chemistry. The VLM assessment is not validated against human expert judgment (e.g., chemists or lab techs).
  - The paper does not demonstrate that agents trained in the generated environments can perform any of the presented protocols. Even without real-world validation, simulated protocol execution (e.g., sequential workstation traversal) could have been evaluated and would have strengthened the paper’s core claims significantly
  - The baseline comparison is not quite fair: Holodeck and SceneWeaver are evaluated on metrics requiring protocol understanding, a capability they were never designed to have and that were absent from their input prompts. This makes it hard to distinguish LabGen’s layout generation capabilities from the mere presence of input data that the baseline systems were not provided with.
  - The authors introduce their own benchmark (LabTouchstone) and evaluate exclusively against it. This is perhaps inevitable given the novelty of the system they are evaluating; there may not exist other suitable evaluations; however, it does mean that their paper presents no independent metrics for LabGen’s quality

Presentation: 3
- Strengths:
  - Well-written paper; clearly explains each of the three core components of LabBuilder. Appendices provide extensive implementation detail supporting reproducibility.
- Weaknesses:
  - The claims of LabBuilder’s capabilities sometimes overreach, particularly in the abstract and introduction. "Rigorous functional semantics" and "functionally valid" suggest that LabBuilder validates not just geometries and boolean safety flags but also detailed semantics such as instrument operation parameters, pipette volumes, fume hood ratings, etc. The paper would benefit from being explicit about these scope boundaries upfront.

Significance: 2
- Strengths:
  - Addresses a novel problem: diverse scene generation for laboratories, with grounding in protocols, semantically-annoted lab equipment, and chemical safety annotations
- Weaknesses:
  - The realism of these environments may be too shallow for training lab robots beyond basic navigation – that is, to train robots to actually execute protocols. This limitation is not well discussed.
  - The paper does not discuss whether diverse scene generation is the right paradigm for labs versus digital twins, which are commonly discussed in the field of self-driving labs. The paper doesn't discuss motivation for why you'd want to generate many diverse lab layouts rather than building a detailed twin of your specific lab. Tying their results to specific gaps in industry adoption of laboratory robots would strengthen the paper.

Originality: 3
- Strengths:
  - Novel application of scene generation to the laboratory domain. Protocol grounding — using structured experimental protocols to drive asset selection, spatial organization, and safety optimization — is a genuinely new idea. LabTouchstone with chemistry-specific safety metrics is also a novel contribution.

---

> ### Author Rebuttal · Authors · 2026-03-31
>
> Dear Reviewer dNaj,
>
> We sincerely thank you for the thoughtful feedback. We are particularly encouraged by your "Weak Accept" recommendation and acknowledgment of our thorough experiments. Below, we address each concern.
>
> ---
>
> ## Q1: Scope of "Functional Validity"
>
> **Comment:** Does LabBuilder verify equipment specifications (temperature, volume) or just layout-level properties?
>
> **Response:**
>
> Our evaluation of functional validity focuses on chemical safety in asset placement, protocol-based navigation (as detailed in the Appendix), and rigorous geometric constraints, alongside assessments of feasibility and visual quality. While the reviewer’s suggestion points toward a highly insightful direction, we consider it to be beyond the primary scope of this paper, which centers on " Protocol-Grounded Scientific Layout Generation". We hope to explore this in future extensions of our work.
>
> ---
>
> ## Q2: Fairness of Baseline Comparison
>
> **Comment:** Holodeck and SceneWeaver were designed for household scenes, not labs, and lack protocol/safety/navigation mechanisms — making the comparison potentially unfair.
>
> **Response:**
>
> As far as we know,no existing method addresses protocol-grounded lab layout generation; Holodeck and SceneWeaver are more capable scene generation baselines available for comparison.Ablation results (Table 3) prove that protocol understanding is the primary differentiator: removing it degrades performance (from 0.700 to 0.300), rendering the model inferior to generic baselines. This gap exists because simple prompts struggle to encode the structured dependencies of a lab, such as: (1) sequential workflows, (2) conditional safety relations.
>
> ---
>
> ## Q3: Poor Geometric Compliance of Baselines
>
> **Comment:** Why do baselines show such poor geometric compliance (e.g., OB=10.8 vs. 0.07) when collision avoidance is a solved geometric problem?
>
> **Response:**
>
> The gap stems from fundamental household-vs-lab differences: (1) **No desktop-level placement understanding** — baselines lack knowledge of correct on-surface object placement, causing frequent boundary violations as a natural consequence; (2) **Missing hierarchical placement** — baselines place assets at floor level, ignoring the room/desktop hierarchy;
>
> ---
>
> ## Q4: Protocol-Level Navigation Evaluation
>
> **Comment:** Could navigation be evaluated at the protocol-execution level rather than just isolated point-goal tasks?
>
> **Response:**
>
> Our navigation planning and execution follow the protocol steps sequentially. When computing the navigation success rate, we aggregate success across all individual steps.We also report the protocol-level navigation success rate below, where a protocol is considered successful only when every step in the sequence is completed.We apologize for any confusion caused by unclear presentation in our manuscript.
>
> |                               | Nav.↑(protocol) |
> | ----------------------------- | :--------------: |
> | Ours (w/o annotation)         |      0.857      |
> | Ours (w/o protocol)           |        -        |
> | Ours (w/o Geom. & Chem. Opt.) |      0.767      |
> | Ours (w/o nav. opt.)          |      0.533      |
> | Ours                          |      0.867      |
>
> ---
>
> ## Q5: Asset Availability Gap
>
> **Comment:** Why is asset availability only 0.833 when protocols are grounded in the asset knowledge base?
>
> **Response:**
>
> Indeed, the protocols are generated from the same asset knowledge base via RAG from natural language experiment descriptions. However, the generation process cannot guarantee complete asset coverage for every protocol — there is inherent generalization that may lead to occasional asset omissions.
>
> ---
>
> ## Q6: VLM-Based Evaluation Reliability
>
> **Comment:** VLM-based evaluation may be unreliable for specialized domains and is not validated against human expert judgment.
>
> **Response:**
>
> We agree that VLM-based evaluation in specialized domains has known reliability issues — a VLM may rate a layout as visually plausible without understanding whether the arrangement actually makes sense for the specified chemistry. VLM scores serve as one of four complementary evaluation dimensions. Our main claims rest on three objective dimensions (Geometric Compliance, FSR, Chemical Safety) — all deterministic/automated. VLM captures subjective visual quality as supplementary evidence.
>
> **Revision**: We will add human expert validation in our work.

---

> > ### Author Rebuttal · Reviewer_dNaj · 2026-04-04
> >
> > I thank the authors for their rebuttal. In response to Q6, appreciate their decision to add human expert validation to supplement their VLM evaluation. My questions Q3-Q6 are resolved.
> >
> > In response to their rebuttal to question 1, I understand that protocol-level validation is out of scope for this paper. However, in that case, I suggest that the authors make this scope statement clearer in their abstract: the phrases "rigorous functional semantics" and "functionally valid" suggest a broader scope that includes functional protocol-level validation, not just safety constraints in scene generation.
> >
> > For question 2, I agree that the ablation study is significant and demonstrates LabBuilder’s value clearly. However, my concern about the informativeness of the baseline comparison remains unaddressed, and I stand by my additional suggestion to make the baselines more fair by providing Holodeck with a more detailed prompt that explicitly lists the required assets and key safety constraints. This would more clearly demonstrate the value-add of LabBuilder over simply using existing scene generation tools with more detailed inputs.
> >
> > The responses to questions 3 and 4 were helpful and clarifying; the hierarchical placement explanation for baseline geometric compliance is convincing, and I am glad to see that protocol-level sequential navigation evaluation exists.

---

> > > ### Author Response · Authors · 2026-04-07
> > >
> > > Thank you for your continued and constructive engagement.
> > >
> > > ---
> > >
> > > ### Q1: Scope Clarification in Abstract
> > >
> > > We agree with this suggestion. The phrases "rigorous functional
> > > semantics" and "functionally valid" in the abstract and introduction
> > > overstate the scope of our current evaluation. We will revise these
> > > to more precisely reflect what LabBuilder verifies: layout-level
> > > functional validity, including protocol-grounded asset selection,
> > > chemical safety constraint satisfaction, geometric compliance, and
> > > navigation reachability. We will add an explicit scope statement
> > > clarifying that equipment specification-level validation (e.g.,
> > > temperature ratings, pipette volumes) and physical manipulation
> > > feasibility are important future directions beyond the current scope.
> > >
> > > ---
> > >
> > > ### Q2: Fairer Baseline Comparison — Holodeck+
> > >
> > > We thank the reviewer for this suggestion and have conducted the
> > > proposed experiment. We introduce **Holodeck+**, which provides
> > > Holodeck with the identical structured protocol used by LabBuilder,
> > > including the full asset list, step-level locations, and chemical
> > > safety constraints, thereby eliminating information asymmetry.
> > >
> > > | Method    | Obj  | OB↓  | CN↓  | Asset↑ | Flam.↑ | Store↑ | Incomp.↑ | Glass↑ |
> > > |-----------|------|------|------|--------|--------|--------|----------|--------|
> > > | Holodeck  | 15.4 | 10.8 | 0.20 | 0.700  | 0.239  | 0.583  | 0.087    | 0.252  |
> > > | Holodeck+ | 19.8 | 15.2 | 0.33 | 0.733  | 0.723  | 0.000  | 0.613    | 0.041  |
> > > | Ours      | 23.2 | 0.07 | 0.17 | 0.833  | 0.725  | 0.801  | 0.716    | 0.364  |
> > >
> > > The results show an informative pattern. On the positive side,
> > > Holodeck+ places more assets (15.4 → 19.8) with better protocol
> > > coverage (Asset 0.700 → 0.733), and distance-based safety metrics
> > > improve notably (Flam. 0.239 → 0.723, Incomp. 0.087 → 0.613). This
> > > confirms that Holodeck can leverage explicit chemical constraints when
> > > provided, and is capable of understanding protocol semantics to some
> > > degree.
> > >
> > > However, this comes with trade-offs. Geometric violations worsen
> > > (OB 10.8 → 15.2, CN 0.20 → 0.33), as placing more assets without
> > > hierarchical placement guidance leads to increased spatial conflicts.
> > > Meanwhile, reagent storage (Store 0.583 → 0.000) and glass edge
> > > avoidance (Glass 0.252 → 0.041) both collapse, suggesting that
> > > Holodeck has no mechanism to enforce placement conventions beyond
> > > what is explicitly stated in the prompt.
> > >
> > > This reveals a fundamental limitation: Holodeck can recognize and
> > > respond to chemical safety cues explicitly stated in the protocol,
> > > but lacks the structured pipeline to simultaneously satisfy geometric
> > > validity, storage conventions, and safety constraints as a whole.
> > > LabBuilder addresses this through hierarchical decomposition,
> > > FastRepair for geometric conflict resolution, and LLMAdjust for
> > > semantic safety reasoning, which together ensure balanced performance
> > > across all dimensions. We will include this comparison in the revised
> > > paper.

---

### Official Review · Reviewer_VSBA · 2026-03-13

**Soundness:** 3
**Presentation:** 3
**Significance:** 2
**Originality:** 4
**Overall Recommendation:** 3
**Confidence:** 3

**Summary:**

This paper presents a scene generation system for generating safe and effective laboratory layouts. Unlike most current scene generation systems, it focuses on functionality. It has 3 parts:
LabForge: Annotated asset and lab protocol data. An LLM is used to structure unstructured data about equipment and lab protocols
LabGen: The generation algorithm. Split into a 2-level hierarchy; the first is "room" level and organizes the big assets (tables, fume-hoods, etc.), while the second "desktop"-level places objects on/in the room assets. This is followed by an iterative optimization with hard geometry and safety constraints.
LabTouchstone: The evaluation framework, which combines traditional scene generation metrics with functionality metrics: can the experimental protocol by performed in the scene, is it safe to do so, and how well can a robot navigate it.

**Compliance With Llm Reviewing Policy:**

Affirmed.

**Final Justification:**

The authors addressed my concerns about missing details and errors analysis in their rebuttal. While reading other reviewer's comments gave me a bit more pause about significance, particularly the gulf between what this paper proposes and what is necessary for real world lab automation, I think that this work is introducing a novel version of scene generation, and it's okay that they haven't quite solved it. I am increasing my recommendation to weak accept, and changing my presentation (+1) and soundness (-1) scores to reflect the author's responses and doubts sown by reading the full review discussion.

**Key Questions For Authors:**

1. There are many missing implementation details. Please give the definitions of pθ, hybrid operator Φ, FastRepair, FastAdjust, and refinement operator Υ. None of these names appear in Appendix C where they are supposed to be detailed.
2. What are the illustrated bounding boxes in Fig 3 pointing to? The qualitative evaluation is difficult to follow since the annotations in this figure are not explained.

I am generally convinced by the novelty and application domain of this paper, a full and satisfactory explanation of these pieces and more illustrative qualitative example would likely remove remaining doubt.

**Limitations:**

yes

**Strengths And Weaknesses:**

Strengths:
- novel to include important considerations of functionality in scene generation
- addresses multiple objectives under hard constraints
- includes code and data for reproducibility!

Weaknesses:
- several components are not detailed in the text (pθ, FastRepair, LLMAdjust)
- qualitative evaluation is difficult to follow due to poorly captioned figure

---

> ### Author Rebuttal · Authors · 2026-03-31
>
> Dear Reviewer VSBA,
>
> We sincerely thank you for the thoughtful, constructive, and encouraging feedback. We are truly grateful for your recognition of the novelty of incorporating functionality into scene generation, the significance of addressing multiple objectives under hard constraints, and the value of releasing code and data for reproducibility. We greatly appreciate your positive assessment of our work's originality (rated excellent) and its potential for generating safe and effective laboratory layouts.
>
>
> ## Q1: Missing Implementation Details
>
> ### Definition 1: $p_\theta$ — LLM-Based Hierarchical Layout Generator
>
> $p_\theta$ is a conditional probabilistic model (Gemini 3.0 Pro) generating layouts in two stages (Eq. 1):
>
> $$\text{Stage 1:} \quad (R, \pi) \sim p_\theta(R, \pi \mid x, P, A)$$
>
> $$\text{Stage 2:} \quad D_s \sim p_\theta(D_s \mid x, P, A, s, R, \pi)$$
>
> where $R$: room-level assets with 6-DoF poses; $\pi$: protocol-to-zone assignments; $x$: experiment description; $P$: structured protocol; $A$: asset knowledge base; $D_s$: desktop layout for surface $s$.
>
> **Flow:** Room-level: LLM receives room dimensions, asset catalog, protocol requirements → outputs JSON of positions/orientations + zoning $\pi$. Desktop-level: for each surface $s$, LLM outputs placements in local coordinate frame (Appendix C.1).
>
> ---
>
> ### Definition 2: $\Phi$ — Hybrid Optimization Operator
>
> $\Phi$ is defined in Eq. 4: $L_{t+1} = \Phi(L_t, P, A)$, combining:
>
> $$\Phi(L_t, P, A) = \text{LLMAdjust}\big(\text{FastRepair}(L_t, P, A),\ P, A\big)$$
>
> FastRepair resolves geometric conflicts; LLMAdjust handles remaining semantic violations. **Acceptance:** accept $L_{t+1}$ if $v(L_{t+1})<v(L_t)$, or if $v(L_{t+1})=v(L_t)$ and $F(L_{t+1})>F(L_t)$.
>
> ---
>
> ### Definition 3: FastRepair — Rule-Based Geometric Solver
>
> Deterministic algorithm; $O(n^2)$ per iteration:
>
> ```
> 1. BOUNDARY REPAIR: For each out-of-bounds object o,
>    translate inward by minimum penetration depth.
> 2. COLLISION REPAIR: For each colliding pair (o₁,o₂),
>    compute MTV via 2D rotated bounding-box intersection;
>    translate o₁ by -ε·n, o₂ by +ε·n (ε = overlap + margin).
> 3. WORKSPACE SYNC: Translate all desktop objects on moved
>    surface w by same displacement (Appendix C.3, Eq. 17).
> 4. ITERATE until no violations remain or max 5 rounds.
> ```
>
> ---
>
> ### Definition 4: LLMAdjust — Semantic Layout Refinement
>
> Handles violations requiring semantic reasoning (chemical safety, workflow proximity, complex rearrangements):
>
> ```
> Input: L (after FastRepair), violation report, P, A
> 1. ENCODE: prompt with current layout, violation diagnostics,
>    chemical safety requirements, optimization level.
> 2. QUERY LLM: outputs JSON commands:
>    move(object, pos), rotate(object, angle), swap(o₁,o₂).
> 3. VALIDATE & APPLY: reject adjustments introducing new
>    hard-constraint violations; apply valid ones.
> ```
>
> ---
>
> ### Definition 5: $\Upsilon$ — Navigation-Aware Refinement Operator
>
> $\Upsilon$ defined in Eq. 6: $L_{t+1}=\Upsilon(L_t,P,A)$ (Appendix C.3, Eqs. 8–29):
>
> ```
> For t = 1 to T:
> 1. Project 3D scene to 2D occupancy grid (with agent dilation).
>    Run A*; classify failures: (i) endpoint invalidity,
>    (ii) boundary violation, (iii) path unreachability (Eq. 5).
> 2. Compute r_t = U_t/N_t × 100% (Eq. 11).
> 3. If U_t = 0: stop (Eq. 12).
> 4. Generate adjustments: rotation fixes (Eqs. 19–26),
>    translations (Eqs. 14–17); deduplicate per (object, direction):
>    a*(o,d) = arg max_{a∈A(o,d)} |Δ(a)| (Eqs. 27–29).
> 5. Apply with workspace synchronization; invoke Φ if needed.
> ```
>
> Convergence typically in 2–3 iterations. **Revision plan:** add all five definitions with pseudocode to appendix; align symbols $\Phi$, FastRepair, LLMAdjust, $\Upsilon$ throughout.
>
> ---
>
> ## Q2: Figure 3 Annotations
>
> Figure 3 has two panels: upper = room-level overview; lower = zoomed local views.
>
> **Revision plan:**
>
> 1. **Redesign** Figure 3 with explicitly labeled annotations and a dedicated legend panel that unambiguously defines all visual elements.
> 2. **Enhanced caption** specifying: (a) the two-part layout structure (upper: room-level global view; lower: zoomed local view); (b) the semantic meaning of each bounding box type (asset location vs. violation region); (c) direct labels on key comparative examples (e.g., "flammable reagent near heat source ✗" vs. "proper separation ✓").
>
> ---
>
> ## Q3: More Qualitative Examples
>
> We will add to the appendix:
>
> (1) layouts for multiple reaction types (substitution, condensation, redox, covering all 7 categories / 30 experiments);
>
> (2) A\*-planned navigation path overlays on 2D occupancy grids;
>
> (3) failure case analysis of baselines (Holodeck, SceneWeaver) with explicit violation annotations.
>
> ---
>
> We believe these revisions fully address the presentation concerns without altering the methodology. Thank you again for your valuable feedback.

---

> > ### Author Rebuttal · Reviewer_VSBA · 2026-04-04
> >
> > Thank you for providing the pseudocode snippets! In addition to these, I think the appendix needs to contain the prompt templates associated with $p\theta$ and the Query LLM step of LLMAdjust.
> >
> > What are the failure case analysis results you intend to add for the baselines?

---

> > > ### Author Response · Authors · 2026-04-07
> > >
> > > Thank you for your continued engagement and constructive follow-up questions.
> > >
> > > ---
> > >
> > > ### Q1: Prompt Templates for pθ and LLMAdjust
> > >
> > > We will add the full prompt templates to the appendix. Below is a summary of their structure:
> > >
> > > **pθ (Hierarchical Layout Generator):**
> > > The room-level prompt provides the LLM with room dimensions, the asset catalog, and the structured protocol, and instructs it to output a JSON of asset positions, orientations, and zone assignments. The desktop-level prompt additionally conditions on the room-level result and the specific surface context, and instructs the LLM to output placements in the local surface coordinate frame.
> > >
> > > **LLMAdjust:**
> > > The prompt provides the current layout state, a structured violation report (including violation type, involved objects, and severity), chemical safety requirements derived from the asset knowledge base, and the current optimization level (room or desktop). The LLM is instructed to output a sequence of JSON commands: `move(object, pos)`, `rotate(object, angle)`, or `swap(o₁, o₂)`. Invalid adjustments that introduce new hard-constraint violations are rejected before application.
> > >
> > > We will include the complete prompt templates with concrete examples in the revised appendix.
> > >
> > > ---
> > >
> > > ### Q2: Failure Case Analysis for Baselines
> > >
> > > We will add qualitative failure case analysis with explicit violation annotations. The analysis reveals four consistent and interpretable failure modes in Holodeck and SceneWeaver:
> > >
> > > **(1) Asset insufficiency.**
> > > Required instruments or reagents are frequently missing or mismatched with the protocol, rendering the layout infeasible for execution. This is reflected quantitatively in low asset availability scores (Holodeck: 0.700, SceneWeaver: 0.226 vs. ours: 0.833).
> > >
> > > **(2) Geometric violations.**
> > > Both baselines exhibit frequent object collisions and out-of-boundary placements (e.g., large furniture extending beyond room boundaries), as confirmed by their OB and CN scores. These violations directly compromise physical realizability.
> > >
> > > **(3) Chemical safety violations.**
> > > Beyond geometric issues, baselines fail to respect basic safety placement rules. This manifests in two distinct ways:
> > > - **(3a)** Arbitrary placement of reagents without respecting storage conventions (e.g., placed on the floor rather than in designated cabinets), most severely reflected in SceneWeaver's storage score of 0.000.
> > > - **(3b)** Failure to maintain safe spatial distances between hazardous items, such as flammable reagents placed near heat sources or incompatible chemicals co-located, due to the absence of chemical knowledge grounding.
> > >
> > > **(4) Workflow and navigation failures.**
> > > As a downstream consequence of (1), layouts generated by baselines lack the asset completeness required to support multi-step protocol execution, making navigation feasibility evaluation inapplicable. This confirms that optimizing for visual plausibility alone is insufficient for protocol-grounded laboratory environments.
> > >
> > > These failure modes are consistent with our central argument that existing methods lack protocol-grounded reasoning and constraint-aware optimization, and we will annotate representative failure cases explicitly in the revised appendix.

---

### Decision · Program_Chairs · 2026-04-30

**Decision:**

Accept (regular)

**Comment:**

The paper addresses a novel and interesting problem; it currently falls short of the technical rigor required, though the AC gives greater credit for the novelty of the setting. On the positive side, reviewers praise the system's architecture, the focus on safety/functionality over aesthetics, and the inclusion of code. However, the paper faces heavy criticism regarding its core claims of "functional validity"; reviewers argue that the protocols are too simplistic (lacking branching or complex logic) and that the VLM-based evaluation lacks validation against human expert judgment. The rebuttal helped clarify some concerns, but the gap between simulation and real-world lab automation remains significant; one reviewer increased the final rating. This is a borderline paper.